# Spectral Inference Networks:
# Unifying Deep and Spectral Learning

**David Pfau**[1], **Stig Petersen**[1], **Ashish Agarwal**[2], **David G. T. Barrett**[1] **& Kimberly L. Stachenfeld**[1]
[1]DeepMind          [2]Google Brain
 London, UK        Mountain View, CA, USA
{pfau, svp, agarwal, barrettdavid, stachenfeld}@google.com

## Abstract

We present Spectral Inference Networks, a framework for learning eigenfunctions of linear operators by stochastic optimization. Spectral Inference Networks generalize Slow Feature Analysis to generic symmetric operators, and are closely related to Variational Monte Carlo methods from computational physics. As such, they can be a powerful tool for unsupervised representation learning from video or graph-structured data. We cast training Spectral Inference Networks as a bilevel optimization problem, which allows for online learning of multiple eigenfunctions. We show results of training Spectral Inference Networks on problems in quantum mechanics and feature learning for videos on synthetic datasets. Our results demonstrate that Spectral Inference Networks accurately recover eigenfunctions of linear operators and can discover interpretable representations from video in a fully unsupervised manner.

## 1    Introduction

Spectral algorithms are central to machine learning and scientific computing. In machine learning, eigendecomposition and singular value decomposition are foundational tools, used for PCA as well as a wide variety of other models. In scientific applications, solving for the eigenfunction of a given linear operator is central to the study of PDEs, and gives the time-independent behavior of classical and quantum systems. For systems where the linear operator of interest can be represented as a reasonably-sized matrix, full eigendecomposition can be achieved in $\mathcal{O}(n^3)$ time (Pan et al., 1998), and in cases where the matrix is too large to diagonalize completely (or even store in memory), iterative algorithms based on Krylov subspace methods can efficiently compute a fixed number of eigenvectors by repeated application of matrix-vector products (Golub & Van Loan, 2012).

At a larger scale, the eigenvectors themselves cannot be represented explicitly in memory. This is the case in many applications in quantum physics and machine learning, where the state space of interest may be combinatorially large or even continuous and high dimensional. Typically, the eigenfunctions of interest are approximated from a fixed number of points small enough to be stored in memory, and then the value of the eigenfunction at other points is approximated by use of the Nyström method (Bengio et al., 2004). As this depends on evaluating a kernel between a new point and every point in the training set, this is not practical for large datasets, and some form of function approximation is necessary. By choosing a function approximator known to work well in a certain domain, such as convolutional neural networks for vision, we may be able to bias the learned representation towards reasonable solutions in a way that is difficult to encode by choice of kernel.

In this paper, we propose a way to approximate eigenfunctions of linear operators on high-dimensional function spaces with neural networks, which we call *Spectral Inference Networks* (SpIN). We show how to train these networks via bilevel stochastic optimization. Our method finds correct eigenfunctions of problems in quantum physics and discovers interpretable representations from video. This significantly extends prior work on unsupervised learning without a generative model and we expect will be useful in scaling many applications of spectral methods.

---

Code is available at https://github.com/deepmind/spectral_inference_networks

The outline of the paper is as follows. Sec 2 provides a review of related work on spectral learning and stochastic optimization of approximate eigenfunctions. Sec. 3 defines the objective function for Spectral Inference Networks, framing eigenfunction problems as an optimization problem. Sec. 4 describes the algorithm for training Spectral Inference Networks using bilevel optimization and a custom gradient to learn ordered eigenfunctions simultaneously. Experiments are presented in Sec. 5 and future directions are discussed in Sec. 6. We also include supplementary materials with more in-depth derivation of the custom gradient updates (Sec. A), a TensorFlow implementation of the core algorithm (Sec. B), and additional experimental results and training details (Sec. C).

## 2   RELATED WORK

Spectral methods are mathematically ubiquitous, arising in a number of diverse settings. Spectral clustering (Ng et al., 2002), normalized cuts (Shi & Malik, 2000) and Laplacian eigenmaps (Belkin & Niyogi, 2002) are all machine learning applications of spectral decompositions applied to graph Laplacians. Related manifold learning algorithms like LLE (Tenenbaum et al., 2000) and IsoMap (Roweis & Saul, 2000) also rely on eigendecomposition, with a different kernel. Spectral algorithms can also be used for asymptotically exact estimation of parametric models like hidden Markov models and latent Dirichlet allocation by computing the SVD of moment statistics (Hsu et al., 2012; Anandkumar et al., 2012).

In the context of reinforcement learning, spectral decomposition of predictive state representations has been proposed as a method for learning a coordinate system of environments for planning and control (Boots et al., 2011), and when the transition function is symmetric its eigenfunctions are also known as *proto-value functions* (PVFs) (Mahadevan & Maggioni, 2007). PVFs have also been proposed by neuroscientists as a model for the emergence of grid cells in the entorhinal cortex (Stachenfeld et al., 2017). The use of PVFs for discovering subgoals in reinforcement learning has been investigated in (Machado et al., 2017) and combined with function approximation in (Machado et al., 2018), though using a less rigorous approach to eigenfunction approximation than SpIN. A qualitative comparison of the two approaches is given in the supplementary material in Sec. C.3.

Spectral learning with stochastic approximation has a long history as well. Probably the earliest work on stochastic PCA is that of "Oja's rule" (Oja, 1982), which is a Hebbian learning rule that converges to the first principal component, and a wide variety of online SVD algorithms have appeared since. Most of these stochastic spectral algorithms are concerned with learning fixed-size eigenvectors from online data, while we are concerned with cases where the eigenfunctions are over a space too large to be represented efficiently with a fixed-size vector.

The closest related work in machine learning on finding eigenfunctions by optimization of parametric models is Slow Feature Analysis (SFA) (Wiskott & Sejnowski, 2002), which is a special case of SpIN. SFA is equivalent to function approximation for Laplacian eigenmaps (Sprekeler, 2011), and it has been shown that optimizing for the slowness of features in navigation can also lead to the emergence of units whose response properties mimic grid cells in the entorhinal cortex of rodents (Wyss et al., 2006; Franzius et al., 2007). SFA has primarily been applied to train shallow or linear models, and when trained on deep models is typically trained in a layer-wise fashion, rather than end-to-end (Kompella et al., 2012; Sun et al., 2014). The features in SFA are learned sequentially, from slowest to fastest, while SpIN allows for simultaneous learning of all eigenfunctions, which is more useful in an online setting.

Spectral methods and deep learning have been combined in other ways. The spectral networks of Bruna et al. (2014) are a generalization of convolutional neural networks to graph and manifold structured data based on the idea that the convolution operator is diagonal in a basis defined by eigenvectors of the Laplacian. In (Ionescu et al., 2015) spectral decompositions were incorporated as differentiable layers in deep network architectures. Spectral decompositions have been used in combination with the kernelized Stein gradient estimator to better learn implicit generative models like GANs (Shi et al., 2018). While these use spectral methods to design or train neural networks, our work uses neural networks to solve large-scale spectral decompositions.

In computational physics, the field of approximating eigenfunctions of a Hamiltonian operator is known as *Variational Quantum Monte Carlo* (VMC) (Foulkes et al., 2001). VMC methods are usually applied to finding the ground state (lowest eigenvalue) of electronic systems, but extensions

to excited states (higher eigenvalues) have been proposed (Blunt et al., 2015). Typically the class of function approximator is tailored to the system, but neural networks have been used for calculating ground states (Carleo & Troyer, 2017) and excited states (Choo et al., 2018). Stochastic optimization for VMC dates back at least to Harju et al. (1997). Most of these methods use importance sampling from a well-chosen distribution to eliminate the bias due to finite batch sizes. In machine learning we are not free to choose the distribution from which the data is sampled, and thus cannot take advantage of these techniques.

## 3 SPECTRAL DECOMPOSITION AS OPTIMIZATION

### 3.1 FINITE-DIMENSIONAL EIGENVECTORS

Eigenvectors of a matrix $\mathbf{A}$ are defined as those vectors $\mathbf{u}$ such that $\mathbf{A}\mathbf{u} = \lambda\mathbf{u}$ for some scalar $\lambda$, the eigenvalue. It is also possible to define eigenvectors as the solution to an optimization problem. If $\mathbf{A}$ is a symmetric matrix, then the largest eigenvector of $\mathbf{A}$ is the solution of:

$$\max_{\substack{\mathbf{u} \\ \mathbf{u}^T\mathbf{u}=1}} \mathbf{u}^T\mathbf{A}\mathbf{u} \tag{1}$$

or equivalently (up to a scaling factor in $\mathbf{u}$)

$$\max_{\mathbf{u}} \frac{\mathbf{u}^T\mathbf{A}\mathbf{u}}{\mathbf{u}^T\mathbf{u}} \tag{2}$$

This is the *Rayleigh quotient*, and it can be seen by setting derivatives equal to zero that this is equivalent to finding $\mathbf{u}$ such that $A\mathbf{u} = \lambda\mathbf{u}$, where $\lambda$ is equal to the value of the Rayleigh quotient. We can equivalently find the lowest eigenvector of $\mathbf{A}$ by minimizing the Rayleigh quotient instead. Amazingly, despite being a nonconvex problem, algorithms such as power iteration converge to the global solution of this problem (Daskalakis et al., 2018, Sec. 4).

To compute the top $N$ eigenvectors $\mathbf{U} = (\mathbf{u}_1, \ldots, \mathbf{u}_N)$, we can solve a sequence of maximization problems:

$$\mathbf{u}_i = \arg\max_{\substack{\mathbf{u} \\ \mathbf{u}_j^T\mathbf{u}=0 \\ j<i}} \frac{\mathbf{u}^T\mathbf{A}\mathbf{u}}{\mathbf{u}^T\mathbf{u}} \tag{3}$$

If we only care about finding a subspace that spans the top $N$ eigenvectors, we can divide out the requirement that the eigenvectors are orthogonal to one another, and reframe the problem as a single optimization problem (Edelman et al., 1998, Sec. 4.4):

$$\max_{\mathbf{U}} \mathrm{Tr}\left((\mathbf{U}^T\mathbf{U})^{-1}\mathbf{U}^T\mathbf{A}\mathbf{U}\right) \tag{4}$$

or, if $\mathbf{u}^i$ denotes row $i$ of $\mathbf{U}$:

$$\max_{\mathbf{U}} \mathrm{Tr}\left(\left(\sum_i \mathbf{u}^{iT}\mathbf{u}^i\right)^{-1} \sum_{ij} A_{ij}\mathbf{u}^{iT}\mathbf{u}^j\right) \tag{5}$$

Note that this objective is invariant to right-multiplication of $\mathbf{U}$ by an arbitrary matrix, and thus we do not expect the columns of $\mathbf{U}$ to be the separate eigenvectors. We will discuss how to break this symmetry in Sec. 4.1.

### 3.2 FROM EIGENVECTORS TO EIGENFUNCTIONS

We are interested in the case where both $\mathbf{A}$ and $\mathbf{u}$ are too large to represent in memory. Suppose that instead of a matrix $\mathbf{A}$ we have a symmetric (not necessarily positive definite) kernel $k(\mathbf{x}, \mathbf{x}')$ where $\mathbf{x}$ and $\mathbf{x}'$ are in some measurable space $\Omega$, which could be either continuous or discrete. Let the inner product on $\Omega$ be defined with respect to a probability distribution with density $p(\mathbf{x})$, so that $\langle f, g \rangle = \int f(\mathbf{x})g(\mathbf{x})p(\mathbf{x})d\mathbf{x} = \mathbb{E}_{\mathbf{x}\sim p(\mathbf{x})}[f(\mathbf{x})g(\mathbf{x})]$. In theory this could be an improper

density, such as the uniform distribution over $\mathbb{R}^n$, but to evaluate it numerically there must be some proper distribution over $\Omega$ from which the data are sampled. We can construct a symmetric linear operator $\mathcal{K}$ from $k$ as $\mathcal{K}[f](\mathbf{x}) = \mathbb{E}_{\mathbf{x}'}[k(\mathbf{x}, \mathbf{x}')f(\mathbf{x}')]$. To compute a function that spans the top $N$ eigenfunctions of this linear operator, we need to solve the equivalent of Eq. 5 for function spaces. Replacing rows $i$ and $j$ with points $\mathbf{x}$ and $\mathbf{x}'$ and sums with expectations, this becomes:

$$\max_{\mathbf{u}} \mathrm{Tr}\left(\mathbb{E}_{\mathbf{x}}\left[\mathbf{u}(\mathbf{x})\mathbf{u}(\mathbf{x})^T\right]^{-1}\mathbb{E}_{\mathbf{x},\mathbf{x}'}\left[k(\mathbf{x}, \mathbf{x}')\mathbf{u}(\mathbf{x})\mathbf{u}(\mathbf{x}')^T\right]\right) \tag{6}$$

where the optimization is over all functions $\mathbf{u} : \Omega \rightarrow \mathbb{R}^N$ such that each element of $\mathbf{u}$ is an integrable function under the metric above. Also note that as $\mathbf{u}^i$ is a row vector while $\mathbf{u}(\mathbf{x})$ is a column vector, the transposes are switched. This is equivalent to solving the constrained optimization problem

$$\max_{\mathbb{E}_{\mathbf{x}}\left[\mathbf{u}(\mathbf{x})\mathbf{u}(\mathbf{x})^T\right]=\mathbf{I}} \mathrm{Tr}\left(\mathbb{E}_{\mathbf{x},\mathbf{x}'}\left[k(\mathbf{x}, \mathbf{x}')\mathbf{u}(\mathbf{x})\mathbf{u}(\mathbf{x}')^T\right]\right) \tag{7}$$

For clarity, we will use $\boldsymbol{\Sigma} = \mathbb{E}_{\mathbf{x}}\left[\mathbf{u}(\mathbf{x})\mathbf{u}(\mathbf{x})^T\right]$ to denote the covariance[1] of features and $\boldsymbol{\Pi} = \mathbb{E}_{\mathbf{x},\mathbf{x}'}\left[k(\mathbf{x}, \mathbf{x}')\mathbf{u}(\mathbf{x})\mathbf{u}(\mathbf{x}')^T\right]$ to denote the kernel-weighted covariance throughout the paper, so the objective in Eq. 6 becomes $\mathrm{Tr}(\boldsymbol{\Sigma}^{-1}\boldsymbol{\Pi})$. The empirical estimate of these quantities will be denoted as $\hat{\boldsymbol{\Sigma}}$ and $\hat{\boldsymbol{\Pi}}$.

## 3.3 KERNELS

The form of the kernel $k$ often allows for simplification to Eq. 6. If $\Omega$ is a graph, and $k(\mathbf{x}, \mathbf{x}') = -1$ if $\mathbf{x} \neq \mathbf{x}'$ and are neighbors and 0 otherwise, and $k(\mathbf{x}, \mathbf{x})$ is equal to the total number of neighbors of $\mathbf{x}$, this is the *graph Laplacian*, and can equivalently be written as:

$$k(\mathbf{x}, \mathbf{x}')\mathbf{u}(\mathbf{x})\mathbf{u}(\mathbf{x}')^T = \left(\mathbf{u}(\mathbf{x}) - \mathbf{u}(\mathbf{x}')\right)\left(\mathbf{u}(\mathbf{x}) - \mathbf{u}(\mathbf{x}')\right)^T \tag{8}$$

for neighboring points (Sprekeler, 2011, Sec. 4.1). It's clear that this kernel penalizes the difference between neighbors, and in the case where the neighbors are adjacent video frames this is Slow Feature Analysis (SFA) (Wiskott & Sejnowski, 2002). Thus SFA is a special case of SpIN, and the algorithm for learning in SpIN here allows for end-to-end online learning of SFA with arbitrary function approximators. The equivalent kernel to the graph Laplacian for $\Omega = \mathbb{R}^n$ is

$$k(\mathbf{x}, \mathbf{x}') = \lim_{\epsilon \to 0} \sum_{i=1}^{n} \epsilon^{-2}(2\delta(\mathbf{x} - \mathbf{x}') - \delta(\mathbf{x} - \mathbf{x}' - \epsilon\mathbf{e}_i) - \delta(\mathbf{x} - \mathbf{x}' + \epsilon\mathbf{e}_i)) \tag{9}$$

where $\mathbf{e}_i$ is the unit vector along the axis $i$. This converges to the *differential* Laplacian, and the linear operator induced by this kernel is $\nabla^2 \triangleq \sum_i \frac{\partial^2}{\partial x_i^2}$, which appears frequently in physics applications. The generalization to generic manifolds is the *Laplace-Beltrami* operator. Since these are purely local operators, we can replace the double expectation over $\mathbf{x}$ and $\mathbf{x}'$ with a single expectation.

## 4 METHOD

There are many possible ways of solving the optimization problems in Equations 6 and 7. In principle, we could use a constrained optimization approach such as the augmented Lagrangian method (Bertsekas, 2014), which has been successfully combined with deep learning for approximating maximum entropy distributions (Loaiza-Ganem et al., 2017). In our experience, such an approach was difficult to stabilize. We could also construct an orthonormal function basis and then learn some flow that preserves orthonormality. This approach has been suggested for quantum mechanics problems by Cranmer et al. (2018). But, if the distribution $p(\mathbf{x})$ is unknown, then the inner product $\langle f, g \rangle$ is not known, and constructing an explicitly orthonormal function basis is not possible. Also, flows can only be defined on continuous spaces, and we are interested in methods that work for large discrete spaces as well. Instead, we take the approach of directly optimizing the quotient in Eq. 6.

---

[1]Technically, this is the second moment, as $\mathbf{u}(\mathbf{x})$ is not necessarily zero-mean, but we will refer to it as the covariance for convenience.

### 4.1 LEARNING ORDERED EIGENFUNCTIONS

Since Eq. 6 is invariant to linear transformation of the features $\mathbf{u}(\mathbf{x})$, optimizing it will only give a function that *spans* the top $N$ eigenfunctions of $\mathcal{K}$. If we were to instead sequentially optimize the Rayleigh quotient for each function $u_i(\mathbf{x})$:

$$\max_{\substack{u_i \\ \mathbb{E}_{\mathbf{x}}[u_i(\mathbf{x})u_i(\mathbf{x})]=1 \\ \mathbb{E}_{\mathbf{x}}[u_i(\mathbf{x})u_j(\mathbf{x})]=0 \\ j=1,\dots,i-1}} \mathbb{E}_{\mathbf{x},\mathbf{x}'} \left[ k(\mathbf{x},\mathbf{x}')u_i(\mathbf{x})u_i(\mathbf{x}') \right] \tag{10}$$

we would recover the eigenfunctions in order. However, this would be cumbersome in an online setting. It turns out that by masking the flow of information from the gradient of Eq. 6 correctly, we can simultaneously learn all eigenfunctions in order.

First, we can use the invariance of trace to cyclic permutation to rewrite the objective in Eq. 6 as $\mathrm{Tr}\left( \mathbf{\Pi}^{-1}\mathbf{\Sigma} \right) = \mathrm{Tr}\left( \mathbf{L}^{-T}\mathbf{L}^{-1}\mathbf{\Sigma} \right) = \mathrm{Tr}\left( \mathbf{L}^{-1}\mathbf{\Pi}\mathbf{L}^{-T} \right)$ where $\mathbf{L}$ is the Cholesky decomposition of $\mathbf{\Sigma}$. Let $\mathbf{\Lambda} = \mathbf{L}^{-1}\mathbf{\Pi}\mathbf{L}^{-T}$, this matrix has the convenient property that the upper left $n \times n$ block only depends on the first $n$ functions $\mathbf{u}_{1:n}(\mathbf{x}) = (u_1(\mathbf{x}),\dots,u_n(\mathbf{x}))^T$. This means the maximum of $\sum_{i=1}^{n}\Lambda_{ii}$ with respect to $\mathbf{u}_{1:n}(\mathbf{x})$ spans the first $n < N$ eigenfunctions. If we additionally mask the gradients of $\Lambda_{ii}$ so they are also independent of any $u_j(\mathbf{x})$ where $j$ is *less than* $i$:

$$\frac{\tilde{\partial}\Lambda_{ii}}{\partial u_j} = \begin{cases} \frac{\partial \Lambda_{ii}}{\partial u_j} & \text{if } i = j \\ 0 & \text{otherwise} \end{cases} \tag{11}$$

and combine the gradients for each $i$ into a single masked gradient $\tilde{\nabla}_{\mathbf{u}}\mathrm{Tr}(\mathbf{\Lambda}) = \sum_i \tilde{\nabla}_{\mathbf{u}}\Lambda_{ii} = \left( \frac{\partial \Lambda_{11}}{\partial u_1}, \dots, \frac{\partial \Lambda_{NN}}{\partial u_N} \right)$ which we use for gradient ascent, then this is equivalent to independently optimizing each $u_i(\mathbf{x})$ towards the objective $\Lambda_{ii}$. Note that there is still nothing forcing all $\mathbf{u}(\mathbf{x})$ to be orthogonal. If we explicitly orthogonalize $\mathbf{u}(\mathbf{x})$ by multiplication by $\mathbf{L}^{-1}$, then we claim that the resulting $\mathbf{v}(\mathbf{x}) = \mathbf{L}^{-1}\mathbf{u}(\mathbf{x})$ will be the true ordered eigenfunctions of $\mathcal{K}$. A longer discussion justifying this is given in the supplementary material in Sec. A. The closed form expression for the masked gradient, also derived in the supplementary material, is given by:

$$\tilde{\nabla}_{\mathbf{u}}\mathrm{Tr}(\mathbf{\Lambda}) = \mathbb{E}[k(\mathbf{x},\mathbf{x}')\mathbf{u}(\mathbf{x})^T]\mathbf{L}^{-T}\mathrm{diag}(\mathbf{L})^{-1} - \mathbb{E}[\mathbf{u}(\mathbf{x})^T]\mathbf{L}^{-T}\mathrm{triu}\left( \mathbf{\Lambda}\mathrm{diag}(\mathbf{L})^{-1} \right) \tag{12}$$

where $\mathrm{triu}$ and $\mathrm{diag}$ give the upper triangular and diagonal of a matrix, respectively. This gradient can then be passed as the error from $\mathbf{u}$ back to parameters $\theta$, yielding:

$$\tilde{\nabla}_{\theta}\mathrm{Tr}(\mathbf{\Lambda}) = \mathbb{E}\left[ k(\mathbf{x},\mathbf{x}')\mathbf{u}(\mathbf{x})^T\mathbf{L}^{-T}\mathrm{diag}(\mathbf{L})^{-1}\frac{\partial \mathbf{u}}{\partial \theta} \right] - \mathbb{E}\left[ \mathbf{u}(\mathbf{x})^T\mathbf{L}^{-T}\mathrm{triu}\left( \mathbf{\Lambda}\mathrm{diag}(\mathbf{L})^{-1} \right)\frac{\partial \mathbf{u}}{\partial \theta} \right] \tag{13}$$

To simplify notation we can express the above as

$$\tilde{\nabla}_{\theta}\mathrm{Tr}(\mathbf{\Lambda}) = \mathbb{E}\left[ \mathbf{J}_{\mathbf{\Pi}}\left( \mathbf{L}^{-T}\mathrm{diag}(\mathbf{L})^{-1} \right) \right] - \mathbb{E}\left[ \mathbf{J}_{\mathbf{\Sigma}}\left( \mathbf{L}^{-T}\mathrm{triu}\left( \mathbf{\Lambda}\mathrm{diag}(\mathbf{L})^{-1} \right) \right) \right] \tag{14}$$

Where $\mathbf{J}_{\mathbf{\Pi}}(\mathbf{A}) = k(\mathbf{x},\mathbf{x}')\mathbf{u}(\mathbf{x})^T\mathbf{A}\frac{\partial \mathbf{u}}{\partial \theta}$ and $\mathbf{J}_{\Sigma}(\mathbf{A}) = \mathbf{u}(\mathbf{x})^T\mathbf{A}\frac{\partial \mathbf{u}}{\partial \theta}$ are linear operators that denote left-multiplication of the Jacobian of $\mathbf{\Pi}$ and $\mathbf{\Sigma}$ with respect to $\theta$ by $\mathbf{A}$. A TensorFlow implementation of this gradient is given in the supplementary material in Sec. B.

### 4.2 BILEVEL OPTIMIZATION

The expression in Eq. 14 is a nonlinear function of multiple expectations, so naively replacing $\mathbf{\Pi}$, $\mathbf{\Sigma}$, $\mathbf{L}$, $\mathbf{\Lambda}$ and their gradients with empirical estimates will be biased. This makes learning Spectral Inference Networks more difficult than standard problems in machine learning for which unbiased gradient estimates are available. We can however reframe this as a *bilevel* optimization problem, for which convergent algorithms exist. Bilevel stochastic optimization is the problem of simultaneously solving two coupled minimization problems $\min_x f(\mathbf{x},\mathbf{y})$ and $\min_y g(\mathbf{x},\mathbf{y})$ for which we only have

---

**Algorithm 1** Learning in Spectral Inference Networks

1: **given** symmetric kernel $k$, decay rates $\beta_t$, first order optimizer OPTIM
2: **initialize** parameters $\theta_0$, average covariance $\bar{\Sigma}_0 = \mathbf{I}$, average Jacobian of covariance $\bar{\mathbf{J}}_{\Sigma_0} = 0$
3: **while** not converged **do**
4:      Get minibatches $\mathbf{x}_{t1}, \ldots, \mathbf{x}_{tN}$ and $\mathbf{x}'_{t1}, \ldots, \mathbf{x}'_{tN}$
5:      $\hat{\Sigma}_t = \frac{1}{2}\left(\frac{1}{N}\sum_i \mathbf{u}_{\theta_t}(\mathbf{x}_{ti})\mathbf{u}_{\theta_t}(\mathbf{x}_{ti})^T + \frac{1}{N}\sum_i \mathbf{u}_{\theta_t}(\mathbf{x}'_{ti})\mathbf{u}_{\theta_t}(\mathbf{x}'_{ti})^T\right)$, covariance of minibatches
6:      $\hat{\Pi}_t = \frac{1}{N}\sum_i k(\mathbf{x}_{ti}, \mathbf{x}'_{ti})\mathbf{u}_{\theta_t}(\mathbf{x}_{ti})\mathbf{u}_{\theta_t}(\mathbf{x}'_{ti})^T$
7:      $\bar{\Sigma}_t \leftarrow (1 - \beta_t)\bar{\Sigma}_{t-1} + \beta_t\hat{\Sigma}_t$
8:      $\bar{\mathbf{J}}_{\Sigma_t} \leftarrow (1 - \beta_t)\bar{\mathbf{J}}_{\Sigma_{t-1}} + \beta_t\hat{\mathbf{J}}_{\Sigma_t}$
9:      $\bar{\mathbf{L}}_t \leftarrow$ Cholesky decomposition of $\bar{\Sigma}_t$
10:     Compute gradient $\tilde{\nabla}_\theta \text{Tr}(\Lambda(\hat{\Pi}_t, \bar{\Sigma}_t, \hat{\mathbf{J}}_{\Pi_t}, \bar{\mathbf{J}}_{\Sigma_t}))$ according to Eq. 14
11:     $\theta_t \leftarrow \text{OPTIM}(\theta_{t-1}, \tilde{\nabla}_\theta\text{Tr}(\Lambda(\hat{\Pi}_t, \bar{\Sigma}_t, \hat{\mathbf{J}}_{\Pi_t}, \bar{\mathbf{J}}_{\Sigma_t})))$
12: **result** Eigenfunctions $\mathbf{v}_{\theta^*}(\mathbf{x}) = \mathbf{L}^{-1}\mathbf{u}_{\theta^*}(\mathbf{x})$ of $\mathcal{K}[f](\mathbf{x}) = \mathbb{E}_{\mathbf{x}'}[k(\mathbf{x}, \mathbf{x}')f(\mathbf{x}')]$

---

noisy unbiased estimates of the gradient of each: $\mathbb{E}[\mathbf{F}(\mathbf{x}, \mathbf{y})] = \nabla_\mathbf{x} f(\mathbf{x}, \mathbf{y})$ and $\mathbb{E}[\mathbf{G}(\mathbf{x}, \mathbf{y})] = \nabla_\mathbf{y} g(\mathbf{x}, \mathbf{y})$. Bilevel stochastic problems are common in machine learning and include actor-critic methods, generative adversarial networks and imitation learning (Pfau & Vinyals, 2016). It has been shown that by optimizing the coupled functions on two timescales then the optimization will converge to simultaneous local minima of $f$ with respect to $\mathbf{x}$ and $g$ with respect to $\mathbf{y}$ (Borkar, 1997):

$$
\begin{aligned}
\mathbf{x}_t &\leftarrow \mathbf{x}_{t-1} - \alpha_t\mathbf{F}(\mathbf{x}_{t-1}, \mathbf{y}_{t-1}) & (15) \\
\mathbf{y}_t &\leftarrow \mathbf{y}_{t-1} - \beta_t\mathbf{G}(\mathbf{x}_t, \mathbf{y}_{t-1}) & (16)
\end{aligned}
$$

where $\lim_{t\to\infty}\frac{\alpha_t}{\beta_t} = 0$, $\sum_t \alpha_t = \sum_t \beta_t = \infty$, $\sum_t \alpha_t^2 < \infty$, $\sum_t \beta_t^2 < \infty$.

By replacing $\Sigma$ and $\mathbf{J}_\Sigma$ with a moving average in Eq. 14, we can cast learning Spectral Inference Networks as exactly this kind of bilevel problem. Throughout the remainder of the paper, let $\hat{\mathbf{X}}_t$ denote the empirical estimate of a random variable $\mathbf{X}$ from the minibatch at time $t$, and let $\bar{\mathbf{X}}_t$ represent the estimate of $\mathbf{X}$ from a moving average, so $\bar{\Sigma}_t$ and $\bar{\mathbf{J}}_{\Sigma_t}$ are defined as:

$$
\begin{aligned}
\bar{\Sigma}_t &\leftarrow \bar{\Sigma}_{t-1} - \beta_t(\bar{\Sigma}_{t-1} - \hat{\Sigma}_t) & (17) \\
\bar{\mathbf{J}}_{\Sigma_t} &\leftarrow \bar{\mathbf{J}}_{\Sigma_{t-1}} - \beta_t(\bar{\mathbf{J}}_{\Sigma_{t-1}} - \hat{\mathbf{J}}_{\Sigma_t}) & (18)
\end{aligned}
$$

This moving average is equivalent to solving

$$
\min_{\Sigma, \mathbf{J}_\Sigma} \frac{1}{2}\left(||\Sigma - \bar{\Sigma}_t||^2 + ||\mathbf{J}_\Sigma - \bar{\mathbf{J}}_{\Sigma_t}||^2\right) \tag{19}
$$

by stochastic gradient descent and clearly has the true $\Sigma$ and $\mathbf{J}_\Sigma$ as a minimum for a fixed $\theta$. Note that Eq. 14 is a *linear* function of $\Pi$ and $\mathbf{J}_\Pi$, so plugging in $\hat{\Pi}_t$ and $\hat{\mathbf{J}}_{\Pi_t}$ gives an unbiased noisy estimate. By also replacing terms that depend on $\Sigma$ and $\mathbf{J}_\Sigma$ with $\bar{\Sigma}_t$ and $\bar{\mathbf{J}}_{\Sigma_t}$, then alternately updating the moving averages and $\theta_t$, we convert the problem into a two-timescale update. Here $\theta_t$ corresponds to $\mathbf{x}_t$, $\bar{\Sigma}_t$ and $\bar{\mathbf{J}}_{\Sigma_t}$ correspond to $\mathbf{y}_t$, $\tilde{\nabla}_\theta\text{Tr}(\Lambda(\hat{\Pi}_t, \bar{\Sigma}_t, \hat{\mathbf{J}}_{\Pi_t}, \bar{\mathbf{J}}_{\Sigma_t}))$ corresponds to $\mathbf{F}(\mathbf{x}_t, \mathbf{y}_t)$ and $(\bar{\Sigma}_{t-1} - \hat{\Sigma}_t, \bar{\mathbf{J}}_{\Sigma_{t-1}} - \hat{\mathbf{J}}_{\Sigma_t})$ corresponds to $\mathbf{G}(\mathbf{x}_t, \mathbf{y}_t)$.

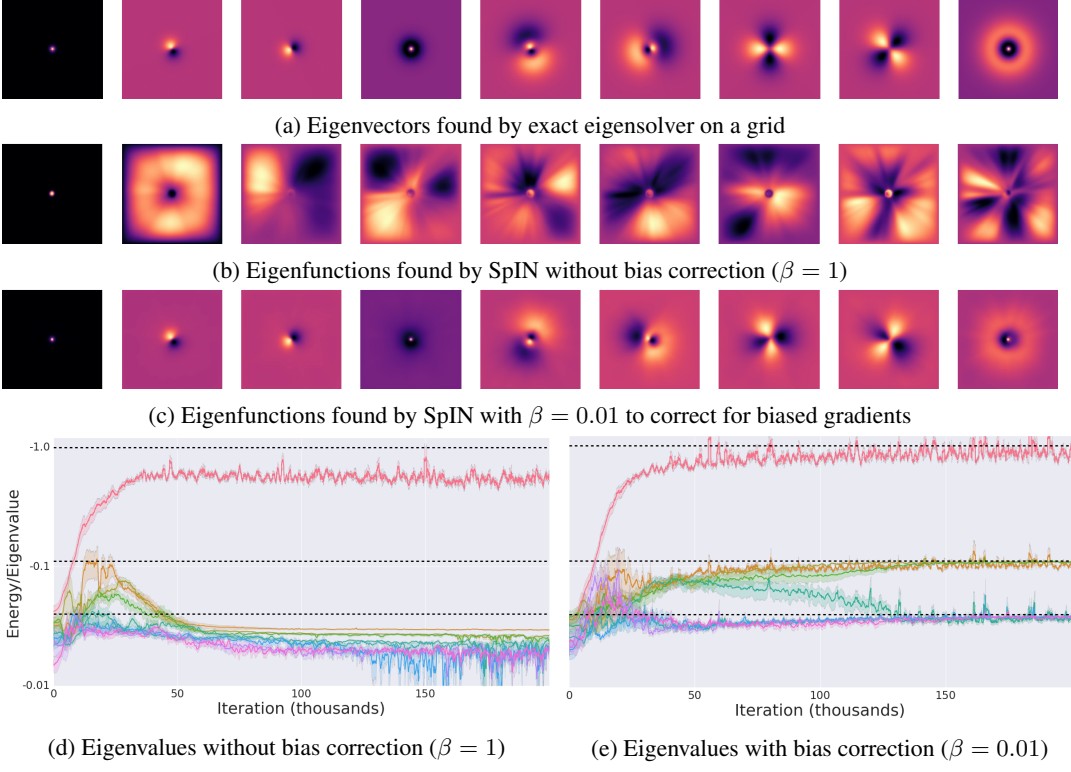

(a) Eigenvectors found by exact eigensolver on a grid

(b) Eigenfunctions found by SpIN without bias correction ($\beta = 1$)

(c) Eigenfunctions found by SpIN with $\beta = 0.01$ to correct for biased gradients

(d) Eigenvalues without bias correction ($\beta = 1$)

(e) Eigenvalues with bias correction ($\beta = 0.01$)

Figure 1: Results of SpIN for solving two-dimensional hydrogen atom.
Black lines in (d) and (e) denote closed-form solution.

## 4.3 Defining Spectral Inference Networks

We can finally combine all these elements together to define what a Spectral Inference Network is. We consider a Spectral Inference Network to be any machine learning algorithm that:

1. Minimizes the objective in Eq. 6 end-to-end by stochastic optimization
2. Performs the optimization over a parametric function class such as deep neural networks
3. Uses the modified gradient in Eq. 14 to impose an ordering on the learned features
4. Uses bilevel optimization to overcome the bias introduced by finite batch sizes

The full algorithm for training Spectral Inference Networks is given in Alg. 1, with TensorFlow pseudocode in the supplementary material in Sec. B. There are two things to note about this algorithm. First, we have to compute an explicit estimate $\hat{\mathbf{J}}_{\boldsymbol{\Sigma}_t}$ of the Jacobian of the covariance with respect to the parameters at each iteration. That means if we have $N$ eigenfunctions we are computing, each step of training will require $N^2$ backward gradient computations. This will be a bottleneck in scaling the algorithm, but we found this approach to be more stable and robust than others. Secondly, while the theory of stochastic optimization depends on proper learning rate schedules, in practice these proper learning rate schedules are rarely used in deep learning. Asymptotic convergence is usually less important than simply getting into the neighborhood of a local minimum, and even for bilevel problems, a careful choice of constant learning rates often suffices for good performance. We follow this practice in our experiments and pick constant values of $\alpha$ and $\beta$.

## 5 Experiments

In this section we present empirical results on a quantum mechanics problem with a known closed-form solution, and an example of unsupervised feature learning from video without a generative

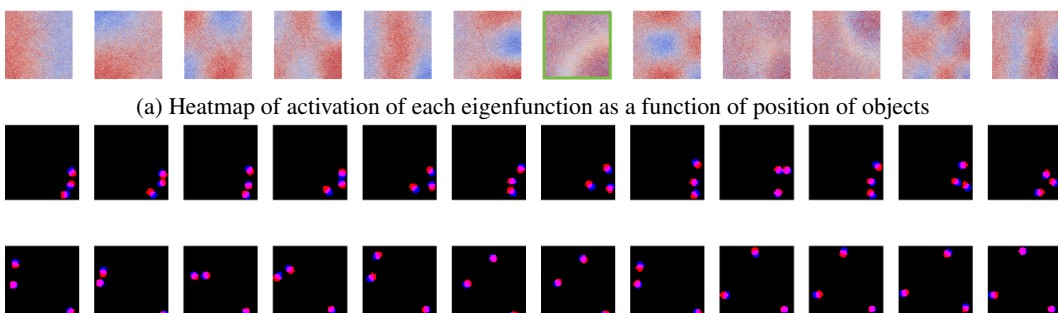

(a) Heatmap of activation of each eigenfunction as a function of position of objects

(b) Frames which most (top) and least (bottom) activate eigenfunction with heatmap outlined in green in Fig. 2a. Successive frames are overlaid in red and blue.

Figure 2: Results of Deep SFA on video of bouncing balls

model. We also provide experiments comparing our approach against the successor feature approach of Machado et al. (2018) for eigenpurose discovery on the Arcade Learning Environment in Sec. C.3 in the supplementary material for the interested reader. Code for the experiments in Sec. 5.1 and C.3 is available at `https://github.com/deepmind/spectral_inference_networks`.

## 5.1 SOLVING THE SCHRÖDINGER EQUATION

As a first experiment to demonstrate the correctness of the method on a problem with a known solution, we investigated the use of SpIN for solving the Schrödinger equation for a two-dimensional hydrogen atom. The time-independent Schrödinger equation for a single particle with mass $m$ in a potential field $V(\mathbf{x})$ is a partial differential equation of the form:

$$E\psi(\mathbf{x}) = \frac{-\hbar^2}{2m}\nabla^2\psi(\mathbf{x}) + V(\mathbf{x})\psi(\mathbf{x}) = \mathcal{H}[\psi](\mathbf{x}) \tag{20}$$

whose solutions describe the wavefunctions $\psi(\mathbf{x})$ with unique energy $E$. The probability of a particle being at position $\mathbf{x}$ then has the density $|\psi(\mathbf{x})|^2$. The solutions are eigenfunctions of the linear operator $\mathcal{H} \triangleq \frac{-\hbar^2}{2m}\nabla^2 + V(\mathbf{x})$ — known as the *Hamiltonian* operator. We set $\frac{\hbar^2}{2m}$ to 1 and choose $V(\mathbf{x}) = \frac{1}{|\mathbf{x}|}$, which corresponds to the potential from a charged particle. In 2 or 3 dimensions this can be solved exactly, and in 2 dimensions it can be shown that there are $2n+1$ eigenfunctions with energy $\frac{-1}{(2n+1)^2}$ for all $n = 0, 1, 2, \ldots$ (Yang et al., 1991).

We trained a standard neural network to approximate the wavefunction $\psi(\mathbf{x})$, where each unit of the output layer was a solution with a different energy $E$. Details of the training network and experimental setup are given in the supplementary material in Sec. C.1. We found it critical to set the decay rate for RMSProp to be slower than the decay $\beta$ used for the moving average of the covariance in SpIN, and expect the same would be true for other adaptive gradient methods. To investigate the effect of biased gradients and demonstrate how SpIN can correct it, we specifically chose a small batch size for our experiments. As an additional baseline over the known closed-form solution, we computed eigenvectors of a discrete approximation to $\mathcal{H}$ on a $128 \times 128$ grid.

Training results are shown in Fig. 1. In Fig. 1a, we see the circular harmonics that make up the electron orbitals of hydrogen in two dimensions. With a small batch size and no bias correction, the eigenfunctions (Fig. 1b) are incorrect and the eigenvalues (Fig. 1d, ground truth in black) are nowhere near the true minimum. With the bias correction term in SpIN, we are able to both accurately estimate the shape of the eigenfunctions (Fig. 1c) and converge to the true eigenvalues of the system (Fig. 1e). Note that, as eigenfunctions 2-4 and 5-9 are nearly degenerate, any linear combination of them is also an eigenfunction, and we do not expect Fig. 1a and Fig. 1c to be identical. The high accuracy of the learned eigenvalues gives strong empirical support for the correctness of our method.

## 5.2 DEEP SLOW FEATURE ANALYSIS

Having demonstrated the effectiveness of SpIN on a problem with a known closed-form solution, we now turn our attention to problems relevant to representation learning in vision. We trained a convolutional neural network to extract features from videos, using the Slow Feature Analysis kernel of Eq. 8. The video is a simple example with three bouncing balls. The velocities of the balls are constant until they collide with each other or the walls, meaning the time dynamics are reversible, and hence the transition function is a symmetric operator. We trained a model with 12 output eigenfunctions using similar decay rates to the experiments in Sec. 5.1. Full details of the training setup are given in Sec. C.2, including training curves in Fig. 3. During the course of training, the order of the different eigenfunctions often switched, as lower eigenfunctions sometimes took longer to fit than higher eigenfunctions.

Analysis of the learned solution is shown in Fig. 2. Fig. 2a is a heatmap showing whether the feature is likely to be positively activated (red) or negatively activated (blue) when a ball is in a given position. Since each eigenfunction is invariant to change of sign, the choice of color is arbitrary. Most of the eigenfunctions are encoding for the position of balls independently, with the first two eigenfunctions discovering the separation between up/down and left/right, and higher eigenfunctions encoding higher frequency combinations of the same thing. However, some eigenfunctions are encoding more complex joint statistics of position. For instance, one eigenfunction (outlined in green in Fig. 2a) has no clear relationship with the marginal position of a ball. But when we plot the frames that most positively or negatively activate that feature (Fig. 2b) we see that the feature is encoding whether all the balls are crowded in the lower right corner, or one is there while the other two are far away. Note that this is a fundamentally *nonlinear* feature, which could not be discovered by a shallow model. Higher eigenfunctions would likely encode for even more complex joint relationships. None of the eigenfunctions we investigated seemed to encode anything meaningful about velocity, likely because collisions cause the velocity to change rapidly, and thus optimizing for slowness of features is unlikely to discover this. A different choice of kernel may lead to different results.

## 6 DISCUSSION

We have shown that a single unified framework is able to compute spectral decompositions by stochastic gradient descent on domains relevant to physics and machine learning. This makes it possible to learn eigenfunctions over very high-dimensional spaces from very large datasets and generalize to new data without the Nyström approximation. This extends work using slowness as a criterion for unsupervised learning without a generative model, and addresses an unresolved issue with biased gradients due to finite batch size. A limitation of the proposed solution is the requirement of computing full Jacobians at every time step, and improving the scaling of training is a promising direction for future research. The physics application presented here is on a fairly simple system, and we hope that Spectral Inference Nets can be fruitfully applied to more complex physical systems for which computational solutions are not yet available. The representations learned on video data show nontrivial structure and sensitivity to meaningful properties of the scene. These representations could be used for many downstream tasks, such as object tracking, gesture recognition, or faster exploration and subgoal discovery in reinforcement learning. Finally, while the framework presented here is quite general, the examples shown investigated only a small number of linear operators. Now that the basic framework has been laid out, there is a rich space of possible kernels and architectures to combine and explore.

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

# Supplementary Material for "Spectral Inference Networks"

## A    BREAKING THE SYMMETRY BETWEEN EIGENFUNCTIONS

Since Eq. 6 is invariant to linear transformation of the features $\mathbf{u}(\mathbf{x})$, optimizing it will only give a function that *spans* the top $K$ eigenfunctions of $\mathcal{K}$. We discuss some of the possible ways to recover ordered eigenfunctions, explain why we chose the approach of using masked gradients, and provide a derivation of the closed form expression for the masked gradient in Eq. 12.

### A.1    ALTERNATIVE STRATEGIES TO BREAK SYMMETRY

If $\mathbf{u}^*(\mathbf{x})$ is a function to $\mathbb{R}^N$ that is an extremum of Eq. 6, then $\mathbb{E}_{\mathbf{x}'}[k(\mathbf{x}, \mathbf{x}')\mathbf{u}^*(\mathbf{x}')] = \mathbf{\Omega}\mathbf{u}^*(\mathbf{x})$ for some matrix $\mathbf{\Omega} \in \mathbb{R}^{N \times N}$ *which is not necessarily diagonal*. We can express this matrix in terms of quantities in the objective:

$$\mathbb{E}_{\mathbf{x}'}[k(\mathbf{x}, \mathbf{x}')\mathbf{u}^*(\mathbf{x}')] = \mathbf{\Omega}\mathbf{u}^*(\mathbf{x}) \tag{21}$$

$$\mathbb{E}_{\mathbf{x}, \mathbf{x}'}\left[k(\mathbf{x}, \mathbf{x}')\mathbf{u}^*(\mathbf{x}')\mathbf{u}^*(\mathbf{x})^T\right] = \mathbf{\Omega}\mathbb{E}_{\mathbf{x}}\left[\mathbf{u}^*(\mathbf{x})\mathbf{u}^*(\mathbf{x})^T\right] \tag{22}$$

$$\mathbf{\Pi} = \mathbf{\Omega}\mathbf{\Sigma} \tag{23}$$

$$\mathbf{\Omega} = \mathbf{\Pi}\mathbf{\Sigma}^{-1} \tag{24}$$

To transform $\mathbf{u}^*(\mathbf{x})$ into ordered eigenfunctions, first we can orthogonalize the functions by multiplying by $\mathbf{L}^{-1}$ where $\mathbf{L}$ is the Cholesky decomposition of $\mathbf{\Sigma}$. Let $\mathbf{v}^*(\mathbf{x}) = \mathbf{L}^{-1}\mathbf{u}^*(\mathbf{x})$, then

$$\mathbb{E}_{\mathbf{x}'}[k(\mathbf{x}, \mathbf{x}')\mathbf{v}^*(\mathbf{x}')] = \mathbf{L}^{-1}\mathbb{E}_{\mathbf{x}'}[k(\mathbf{x}, \mathbf{x}')\mathbf{u}^*(\mathbf{x}')] = \mathbf{L}^{-1}\mathbf{\Pi}\mathbf{\Sigma}^{-1}\mathbf{u}^*(\mathbf{x}) = \mathbf{L}^{-1}\mathbf{\Pi}\mathbf{L}^{-T}\mathbf{v}^*(\mathbf{x})$$

The matrix $\mathbf{L}^{-1}\mathbf{\Pi}\mathbf{L}^{-T} = \mathbf{\Lambda}$ is symmetric, so we can diagonalize it: $\mathbf{\Lambda} = \mathbf{V}\mathbf{D}\mathbf{V}^T$, and then $\mathbf{w}^*(\mathbf{x}) = \mathbf{V}^T\mathbf{v}^*(\mathbf{x}) = \mathbf{V}^T\mathbf{L}^{-1}\mathbf{u}^*(\mathbf{x})$ are true eigenfunctions, with eigenvalues along the diagonal of $\mathbf{D}$. In principle, we could optimize Eq. 6, accumulating statistics on $\mathbf{\Pi}$ and $\mathbf{\Sigma}$, and transform the functions $\mathbf{u}^*$ into $\mathbf{w}^*$ at the end. In practice, we found that the extreme eigenfunctions were "contaminated" by small numerical errors in the others eigenfunctions, and that this approach struggled to learn degenerate eigenfunctions. This inspired us to explore the masked gradient approach instead, which improves numerical robustness.

### A.2    CORRECTNESS OF MASKED GRADIENT APPROACH

Throughout this section, let $\mathbf{x}_{i:j}$ be the slice of a vector from row $i$ to $j$ and let $\mathbf{A}_{i:j,k:\ell}$ be the block of a matrix $\mathbf{A}$ containing rows $i$ through $j$ and columns $k$ through $\ell$. Let $\mathbf{\Sigma} = \mathbb{E}_{\mathbf{x}}[\mathbf{u}(\mathbf{x})\mathbf{u}(\mathbf{x})^T]$, $\mathbf{\Pi} = \mathbb{E}_{\mathbf{x}, \mathbf{x}'}[k(\mathbf{x}, \mathbf{x}')\mathbf{u}(\mathbf{x})\mathbf{u}(\mathbf{x}')^T]$, $\mathbf{L}$ be the Cholesky decomposition of $\mathbf{\Sigma}$ and $\mathbf{\Lambda} = \mathbf{L}^{-1}\mathbf{\Pi}\mathbf{L}^{-T}$. The arguments here are not meant as mathematically rigorous proofs but should give the reader enough of an understanding to be confident that the numerics of our method are correct for optimization over a sufficiently expressive class of functions.

**Claim 1.** $\mathbf{\Lambda}_{1:n,1:n}$ *is independent of* $\mathbf{u}_{n+1:n}(\mathbf{x})$.

The Cholesky decomposition of a positive-definite matrix is the unique lower triangular matrix with positive diagonal such that $\mathbf{L}\mathbf{L}^T = \mathbf{\Sigma}$. Expanding this out into blocks yields:

$$\begin{pmatrix} \mathbf{L}_{1:n,1:n} & 0 \\ \mathbf{L}_{n+1:N,1:n} & \mathbf{L}_{n+1:N,n+1:N} \end{pmatrix} \begin{pmatrix} \mathbf{L}_{1:n,1:n}^T & \mathbf{L}_{n+1:N,1:n}^T \\ 0 & \mathbf{L}_{n+1:N,n+1:N}^T \end{pmatrix} =$$

$$\begin{pmatrix} \mathbf{L}_{1:n,1:n}\mathbf{L}_{1:n,1:n}^T & \mathbf{L}_{1:n,1:n}\mathbf{L}_{n+1:N,1:n}^T \\ \mathbf{L}_{n+1:N,1:n}\mathbf{L}_{1:n,1:n}^T & \mathbf{L}_{n+1:N,1:n}\mathbf{L}_{n+1:N,1:n}^T + \mathbf{L}_{n+1:N,n+1:N}\mathbf{L}_{n+1:N,n+1:N}^T \end{pmatrix} =$$

$$\begin{pmatrix} \mathbf{\Sigma}_{1:n,1:n} & \mathbf{\Sigma}_{1:n,n+1:N} \\ \mathbf{\Sigma}_{n+1:N,1:n} & \mathbf{\Sigma}_{n+1:N,n+1:N} \end{pmatrix}$$

Inspecting the upper left block, we see that $\mathbf{L}_{1:n,1:n}\mathbf{L}_{1:n,1:n}^T = \mathbf{\Sigma}_{1:n,1:n}$. As $\mathbf{L}_{1:n,1:n}$ is also lower-triangular, it must be the Cholesky decomposition of $\mathbf{\Sigma}_{1:n,1:n}$. The inverse of a lower triangular matrix will also be lower triangular, and a similar argument to the one above shows that the upper left block of the inverse of a lower triangular matrix will be the inverse of the upper left block, so the upper left block of $\mathbf{\Lambda}$ can be written as:

$$\mathbf{\Lambda}_{1:n,1:n} = \mathbf{L}_{1:n,1:n}^{-1}\mathbf{\Pi}_{1:n,1:n}\mathbf{L}_{1:n,1:n}^{-T}$$

which depends only on $\mathbf{u}_{1:n}(\mathbf{x})$ □

**Claim 2.** *Let* $\tilde{\nabla}_{\mathbf{u}}\text{Tr}(\mathbf{\Lambda}) = \left(\frac{\partial \Lambda_{11}}{\partial u_1}, \ldots, \frac{\partial \Lambda_{NN}}{\partial u_N}\right)$, *and let* $\mathbf{v}(\mathbf{x}) = \mathbf{L}^{-1}\mathbf{u}(\mathbf{x})$. *If the parameters* $\theta$ *of* $\mathbf{u}(\mathbf{x})$ *are maximized by gradient ascent so that* $\tilde{\nabla}_{\theta}\text{Tr}(\mathbf{\Lambda}) = 0$, *and the true eigenfunctions are in the class of functions parameterized by* $\theta$, *then* $\mathbf{v}(\mathbf{x})$ *will be the eigenfunctions of the operator* $\mathcal{K}$ *defined as* $\mathcal{K}[f](\mathbf{x}) = \mathbb{E}_{\mathbf{x}'}[k(\mathbf{x}, \mathbf{x}')f(\mathbf{x}')]$, *ordered from highest eigenvalue to lowest.*

The argument proceeds by induction. $\Lambda_{11} = L_{11}^{-1}\Pi_{11}L_{11}^{-1} = \frac{\Pi_{11}}{\Sigma_{11}}$, which is simply the Rayleigh quotient in Eq. 10 for $i = 1$. The maximum of this is clearly proportional to the top eigenfunction, and $v_1(\mathbf{x}) = L_{11}^{-1}u_1(\mathbf{x}) = \Sigma_{11}^{-1/2}u_1(\mathbf{x})$ is the normalized eigenfunction.

Now suppose $\mathbf{v}_{1:n}(\mathbf{x}) = \mathbf{L}_{1:n,1:n}^{-1}\mathbf{u}_{1:n}(\mathbf{x})$ are the first $n$ eigenfunctions of $\mathcal{K}$. Because $\mathbf{u}_{1:n}(\mathbf{x})$ span the first $n$ eigenfunctions, and $\Lambda_{ii}$ is independent of $u_{n+1}(\mathbf{x})$ for $i < n+1$, $\mathbf{u}_{1:n+1}(\mathbf{x})$ is a maximum of $\sum_{i=1}^n \Lambda_{ii}$ no matter what the function $u_{n+1}(\mathbf{x})$ is. Training $u_{n+1}(\mathbf{x})$ with the masked gradient $\tilde{\nabla}_{\mathbf{u}}\text{Tr}(\mathbf{\Lambda})$ is equivalent to maximizing $\Lambda_{(n+1)(n+1)}$, so for the optimal $u_{n+1}(\mathbf{x})$, $\mathbf{u}_{1:n+1}(\mathbf{x})$ will be a maximum of $\sum_{i=1}^{n+1} \Lambda_{ii}$. Therefore $\mathbf{u}_{1:n}(\mathbf{x})$ span the first $n$ eigenfunctions and $\mathbf{u}_{1:n+1}(\mathbf{x})$ span the first $n + 1$ eigenfunctions, so orthogonalizing $\mathbf{u}_{1:n+1}(\mathbf{x})$ by multiplication by $\mathbf{L}_{1:n+1,1:n+1}^{-1}$ will subtract anything in the span of the first $n$ eigenfunctions off of $u_{n+1}(\mathbf{x})$, meaning $v_{n+1}(\mathbf{x})$ will be the $(n + 1)$th eigenfunction of $\mathcal{K}$ □

### A.3 DERIVATION OF MASKED GRADIENT

The derivative of the normalized features with respect to parameters can be expressed as

$$\frac{\partial \Lambda_{kk}}{\partial \theta} = \frac{\partial \Lambda_{kk}}{\partial \mathbf{u}}\frac{\partial \mathbf{u}}{\partial \theta} = \left(\frac{\partial \Lambda_{kk}}{\partial \text{vec}(\mathbf{L})}\frac{\partial \text{vec}(\mathbf{L})}{\partial \text{vec}(\mathbf{\Sigma})}\frac{\partial \text{vec}(\mathbf{\Sigma})}{\partial \mathbf{u}} + \frac{\partial \Lambda_{kk}}{\partial \text{vec}(\mathbf{\Pi})}\frac{\partial \text{vec}(\mathbf{\Pi})}{\partial \mathbf{u}}\right)\frac{\partial \mathbf{u}}{\partial \theta} \quad (25)$$

if we flatten out the matrix-valued $\mathbf{L}$, $\mathbf{\Sigma}$ and $\mathbf{\Pi}$.

The reverse-mode sensitivities for the matrix inverse and Cholesky decomposition are given by $\bar{\mathbf{A}} = -\mathbf{C}^T\bar{\mathbf{C}}\mathbf{C}^T$ where $\mathbf{C} = \mathbf{A}^{-1}$ and $\bar{\mathbf{\Sigma}} = \mathbf{L}^{-T}\mathbf{\Phi}(\mathbf{L}^T\bar{\mathbf{L}})\mathbf{L}^{-1}$ where $\mathbf{L}$ is the Cholesky decomposition of $\mathbf{\Sigma}$ and $\mathbf{\Phi}(\cdot)$ is the operator that replaces the upper triangular of a matrix with its lower triangular transposed (Giles, 2008; Murray, 2016). Using this, we can compute the gradients in closed form by application of the chain rule.

To simplify notation slightly, let $\mathbf{\Delta}^k$ and $\mathbf{\Phi}^k$ be matrices defined as:

$$\Delta_{ij}^k = \begin{cases} 1 & \text{if } i = k \text{ and } j = k \\ 0 & \text{otherwise} \end{cases} \qquad \Phi_{ij}^k = \begin{cases} 1 & \text{if } i = k \text{ and } j \leq k \\ 1 & \text{if } i \leq k \text{ and } j = k \\ 0 & \text{otherwise} \end{cases} \quad (26)$$

Then the unmasked gradient has the form:

$$\nabla_{\mathbf{\Pi}}\Lambda_{kk} = \mathbf{L}^{-T}\mathbf{\Delta}^k\mathbf{L}^{-1} \quad (27)$$

$$\nabla_{\mathbf{\Sigma}}\Lambda_{kk} = -\mathbf{L}^{-T}(\mathbf{\Phi}^k \circ \mathbf{\Lambda})\mathbf{L}^{-1} \quad (28)$$

while the gradients of $\mathbf{\Pi}$ and $\mathbf{\Sigma}$ with respect to $\mathbf{u}$ are given (elementwise) by:

$$\frac{\partial \Pi_{ij}}{\partial u_k} = \frac{\partial \mathbb{E}[k(\mathbf{x}, \mathbf{x}')u_i(\mathbf{x})u_j(\mathbf{x}')]}{\partial u_k} = \delta_{ik}\mathbb{E}[k(\mathbf{x}, \mathbf{x}')u_j(\mathbf{x}')] + \delta_{jk}\mathbb{E}[k(\mathbf{x}, \mathbf{x}')u_i(\mathbf{x})] \quad (29)$$

$$\frac{\partial \Sigma_{ij}}{\partial u_k} = \frac{\partial \mathbb{E}[u_i(\mathbf{x})u_j(\mathbf{x})]}{\partial u_k} = \delta_{ik}\mathbb{E}[u_j(\mathbf{x})] + \delta_{jk}\mathbb{E}[u_i(\mathbf{x})] \quad (30)$$

which, in combination, give the unmasked gradient with respect to $\mathbf{u}$ as:

$$\nabla_{\mathbf{u}}\Lambda_{kk} = \sum_{ij} \frac{\partial \Lambda_{kk}}{\partial \Pi_{ij}} \nabla_{\mathbf{u}}\Pi_{ij} + \frac{\partial \Lambda_{kk}}{\partial \Sigma_{ij}} \nabla_{\mathbf{u}}\Sigma_{ij} \quad (31)$$

$$\propto \mathbb{E}[k(\mathbf{x}, \mathbf{x}')\mathbf{u}(\mathbf{x})^T]\nabla_{\mathbf{\Pi}}\Lambda_{kk} + \mathbb{E}[\mathbf{u}(\mathbf{x})^T]\nabla_{\mathbf{\Sigma}}\Lambda_{kk} \quad (32)$$

Here the gradient is expressed as a row vector, to be consistent with Eq. 25, and a factor of 2 has been dropped in the last line that can be absorbed into the learning rate.

To zero out the relevant elements of the gradient $\nabla_{\mathbf{u}}\Lambda_{kk}$ as described in Eq. 11, we can right-multiply by $\mathbf{\Delta}^k$. The masked gradients can be expressed in closed form as:

$$\tilde{\nabla}_{\mathbf{\Pi}}\mathrm{Tr}(\mathbf{\Lambda}) = \sum_k \nabla_{\mathbf{\Pi}}\Lambda_{kk}\mathbf{\Delta}^k = \mathbf{L}^{-T}\mathrm{diag}(\mathbf{L})^{-1} \quad (33)$$

$$\tilde{\nabla}_{\mathbf{\Sigma}}\mathrm{Tr}(\mathbf{\Lambda}) = \sum_k \nabla_{\mathbf{\Sigma}}\Lambda_{kk}\mathbf{\Delta}^k = -\mathbf{L}^{-T}\mathrm{triu}\left(\mathbf{\Lambda}\mathrm{diag}(\mathbf{L})^{-1}\right) \quad (34)$$

$$\tilde{\nabla}_{\mathbf{u}}\mathrm{Tr}(\mathbf{\Lambda}) = \sum_k \mathbb{E}[k(\mathbf{x}, \mathbf{x}')\mathbf{u}(\mathbf{x})^T]\nabla_{\mathbf{\Pi}}\Lambda_{kk}\mathbf{\Delta}^k + \mathbb{E}[\mathbf{u}(\mathbf{x})^T]\nabla_{\mathbf{\Sigma}}\Lambda_{kk}\mathbf{\Delta}^k$$

$$= \mathbb{E}[k(\mathbf{x}, \mathbf{x}')\mathbf{u}(\mathbf{x})^T]\mathbf{L}^{-T}\mathrm{diag}(\mathbf{L})^{-1} - \mathbb{E}[\mathbf{u}(\mathbf{x})^T]\mathbf{L}^{-T}\mathrm{triu}\left(\mathbf{\Lambda}\mathrm{diag}(\mathbf{L})^{-1}\right) \quad (35)$$

where triu and diag give the upper triangular and diagonal of a matrix, respectively. A TensorFlow implementation of this masked gradient is given below.

## B  TENSORFLOW IMPLEMENTATION OF SPIN UPDATE

Here we provide a short pseudocode implementation of the updates in Alg. 1 in TensorFlow. The code is not intended to run as is, and leaves out some global variables, proper initialization and code for constructing networks and kernels. However all nontrivial elements of the updates are given in detail here.

```
import tensorflow as tf
from tensorflow.python.ops.parallel_for import jacobian

@tf.custom_gradient
def covariance(x, y):
  batch_size = float(x.shape[0].value)
  cov = tf.matmul(x, y, transpose_a=True) / batch_size
  def gradient(grad):
    return (tf.matmul(y, grad) / batch_size,
            tf.matmul(x, grad) / batch_size)
  return cov, gradient

@tf.custom_gradient
def eigenvalues(sigma, pi):
  """Eigenvalues as custom op so that we can overload gradients."""
  chol = tf.cholesky(sigma)
  choli = tf.linalg.inv(chol)

  rq = tf.matmul(choli, tf.matmul(pi, choli, transpose_b=True))
```

```
  eigval = tf.matrix_diag_part(rq)
  def gradient(_):
    """Symbolic form of the masked gradient."""
    dl = tf.diag(tf.matrix_diag_part(choli))
    triu = tf.matrix_band_part(tf.matmul(rq, dl), 0, -1)
    dsigma = -1.0*tf.matmul(choli, triu, transpose_a=True)
    dpi = tf.matmul(choli, dl, transpose_a=True)

    return dsigma, dpi
  return eigval, gradient

def moving_average(x, c):
  """Creates moving average operation.

  This is pseudocode for clarity!
  Should actually initialize sigma_avg with tf.eye,
  and should add handling for when x is a list.
  """
  ma = tf.Variable(tf.zeros_like(x), trainable=False)
  ma_update = tf.assign(ma, (1-c)*ma + c*x)
  return ma, ma_update

def spin(x1, x2, network, kernel, params, optim):
  """Function to create TensorFlow ops for learning in SpIN.

  Args:
    x1: first minibatch, of shape (batch size, input dimension)
    x2: second minibatch, of shape (batch size, input dimension)
    network: function that takes minibatch and parameters and
             returns output of neural network
    kernel: function that takes two minibatches and returns
            symmetric function of the inputs
    params: list of tf.Variables with network parameters
    optim: an instance of a tf.train.Optimizer object

  Returns:
    step: op that implements one iteration of SpIN training update
    eigenfunctions: op that gives ordered eigenfunctions
  """

  # `u1` and `u2` are assumed to have the batch elements along first
  # dimension and different eigenfunctions along the second dimension
  u1 = network(x1, params)
  u2 = network(x2, params)

  sigma = 0.5 * (covariance(u1, u1) + covariance(u2, u2))
  sigma.set_shape((u1.shape[1], u1.shape[1]))

  # For the hydrogen examples in Sec. 4.1, `kernel(x1, x2)*u2`
  # can be replaced by the result of applying the operator
  # H to the function defined by `network(x1, params)`.
  pi = covariance(u1, kernel(x1, x2)*u2)
  pi.set_shape((u1.shape[1], u1.shape[1]))

  sigma_jac = jacobian(sigma, params)
  sigma_avg, update_sigma = moving_average(sigma, beta)
  sigma_jac_avg, update_sigma_jac = moving_average(sigma_jac, beta)
```

```
with tf.control_dependencies(update_sigma_jac + [update_sigma]):
  eigval = eigenvalues(sigma_avg, pi)
  loss = tf.reduce_sum(eigval)
  sigma_back = tf.gradients(loss, sigma_avg)[0]

  gradients = [
    tf.tensordot(sigma_back, sig_jac, [[0, 1], [0, 1]]) + grad
    for sig_jac, grad in zip(sigma_jac_avg, tf.gradients(loss, params))
  ]

step = optim.apply_gradients(zip(gradients, params))
eigenfunctions = tf.matmul(u1,
                           tf.linalg.inv(tf.cholesky(sigma_avg)),
                           transpose_b=True)
return step, eigenfunctions
```

## C  EXPERIMENTAL DETAILS

### C.1  SOLVING THE SCHRÖDINGER EQUATION

To solve for the eigenfunctions with lowest eigenvalues, we used a neural network with 2 inputs (for the position of the particle), 4 hidden layers each with 128 units, and 9 outputs, corresponding to the first 9 eigenfunctions. We used a batch size of 128 - much smaller than the 16,384 nodes in the 2D grid used for the exact eigensolver solution. We chose a softplus nonlinearity $\log(1+\exp(x))$ rather than the more common ReLU, as the Laplacian operator $\nabla^2$ would be zero almost everywhere for a ReLU network. We used RMSProp (Tieleman & Hinton, 2012) with a decay rate of 0.999 and learning rate of 1e-5 for all experiments. We sampled points uniformly at random from the box $[-D, D]^2$ during training, and to prevent degenerate solutions due to the boundary condition, we multiplied the output of the network by $\prod_i(\sqrt{2D^2 - x_i^2} - D)$, which forces the network output to be zero at the boundary without the derivative of the output blowing up. We chose $D = 50$ for the experiments shown here. We use the finite difference approximation of the differential Laplacian given in Sec. 3.2 with $\epsilon$ some small number (around 0.1), which takes the form:

$$\nabla^2\psi(\mathbf{x}) \approx \frac{1}{\epsilon^2}\sum_i \psi(\mathbf{x} + \epsilon\mathbf{e}_i) + \psi(\mathbf{x} - \epsilon\mathbf{e}_i) - 2\psi(\mathbf{x}) \tag{36}$$

when applied to $\psi(\mathbf{x})$. Because the Hamiltonian operator is a purely local function of $\psi(\mathbf{x})$, we don't need to sample pairs of points $\mathbf{x}, \mathbf{x}'$ for each minibatch, which simplifies calculations.

We made one additional modification to the neural network architecture to help separation of different eigenfunctions. Each layer had a block-sparse structure that became progressively more separated the deeper into the network it was. For layer $\ell$ out of $L$ with $m$ inputs and $n$ outputs, the weight $w_{ij}$ was only nonzero if there exists $k \in \{1, \ldots, K\}$ such that $i \in [\frac{k-1}{K-1}\frac{\ell-1}{L}m, \frac{k-1}{K-1}\frac{\ell-1}{L}m + \frac{L-\ell+1}{L}m]$ and $j \in [\frac{k-1}{K-1}\frac{\ell-1}{L}n, \frac{k-1}{K-1}\frac{\ell-1}{L}n + \frac{L-\ell+1}{L}n]$. This split the weight matrices into overlapping blocks, one for each eigenfunction, allowing features to be shared between eigenfunctions in lower layers of the network while separating out features which were distinct between eigenfunctions higher in the network.

### C.2  DEEP SLOW FEATURE ANALYSIS

We trained on 200,000 64×64 pixel frames, and used a network with 3 convolutional layers, each with 32 channels, 5×5 kernels and stride 2, and a single fully-connected layer with 128 units before outputting 12 eigenfunctions. We also added a constant first eigenfunction, since the first eigenfunction of the Laplacian operator is always constant with eigenvalue zero. This is equivalent to forcing the features to be zero-mean. We used the same block-sparse structure for the weights that was used in the Schrödinger equation experiments, with sparsity in weights between units extended to sparsity in weights between entire feature maps for the convolutional layers. We trained with RMSProp with learning rate 1e-6 and decay 0.999 and covariance decay rate $\beta = 0.01$ for 1,000,000 iterations.

To make the connection to gradient descent clearer, we use the opposite convention to RMSProp: $\beta = 1$ corresponds to zero memory for the moving average, meaning the RMS term in RMSProp decays ten times more slowly than the covariance moving average in these experiments. Each batch contained 24 clips of 10 consecutive frames. So that the true state was fully observable, we used two consecutive frames as the input $\mathbf{x}_t, \mathbf{x}_{t+1}$ and trained the network so that the difference from that and the features for the frames $\mathbf{x}_{t+1}, \mathbf{x}_{t+2}$ were as small as possible.

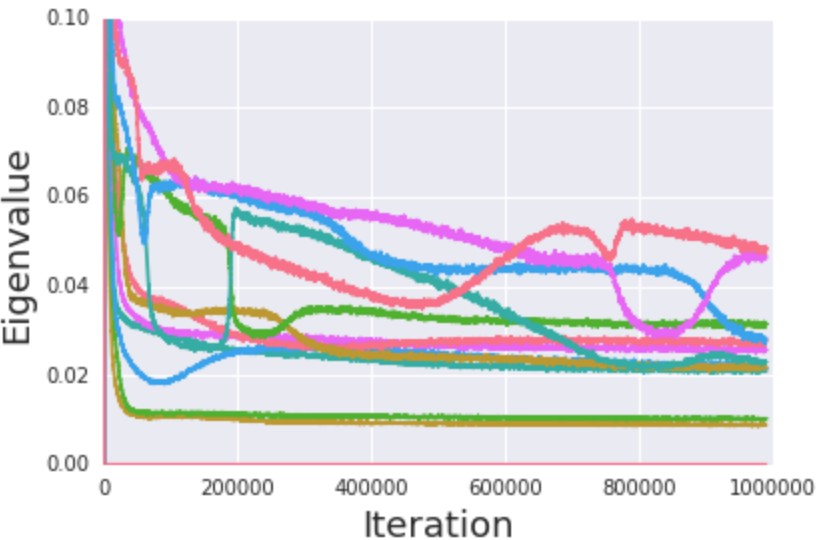

Figure 3: Training curves on bouncing ball videos

## C.3 SUCCESSOR FEATURES AND THE ARCADE LEARNING ENVIRONMENT

We provide a qualitative comparison of the performance of SpIN with the SFA objective against the successor feature approach for learning eigenpurposes Machado et al. (2018) on the Arcade Learning Environment (Bellemare et al., 2013). As in Machado et al. (2018), we trained a network to perform next-frame prediction on 500k frames of a random agent playing one game. We simultaneously trained another network to compute the successor features (Barreto et al., 2017) of the latent code of the next-frame predictor, and computed the "eigenpurposes" by applying PCA to the successor features on 64k held-out frames of gameplay. We used the same convolutional network architecture as Machado et al. (2018), a batch size of 32 and RMSProp with a learning rate of 1e-4 for 300k iterations, and updated the target network every 10k iterations. While the original paper did not mean-center the successor features when computing eigenpurposes, we found that the results were significantly improved by doing so. Thus the baseline presented here is actually stronger than in the original publication.

On the same data, we trained a spectral inference network with the same architecture as the encoder of the successor feature network, except for the fully connected layers, which had 128 hidden units and 5 non-constant eigenfunctions. We tested SpIN on the same 64k held-out frames as those used to estimate the eigenpurposes. We used the same training parameters and kernel as in Sec. 5.2. As SpIN is not a generative model, we must find another way to compare the features learned by each method. We averaged together the 100 frames from the test set that have the largest magnitude positive or negative activation for each eigenfunction/eigenpurpose. Results are shown in Fig. 4, with more examples and comparison against PCA on pixels at the end of this section.

By comparing the top row to the bottom row in each image, we can judge whether that feature is encoding anything nontrivial. If the top and bottom row are noticeably different, this is a good indication that something is being learned. It can be seen that for many games, successor features may find a few eigenpurposes that encode interesting features, but many eigenpurposes do not seem to encode anything that can be distinguished from the mean image. Whereas for SpIN, nearly all eigenfunctions are encoding features such as the presence/absence of a sprite, or different arrange-

Successor Features          Spectral Inference Networks

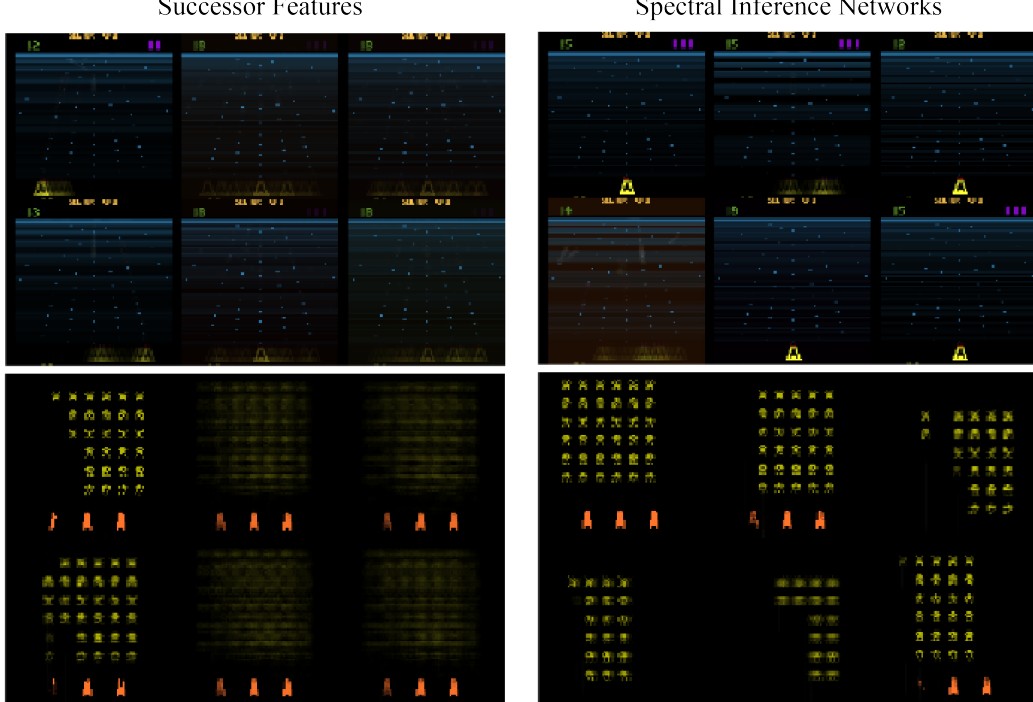

Figure 4: Comparison of Successor Features and SpIN on Beam Rider (top) and Space Invaders (bottom).

ments of sprites, that lead to a clear distinction between the top and bottom row. Moreover, SpIN is able to learn to encode these features in a fully end-to-end fashion, without any pixel reconstruction loss, whereas the successor features must be trained from two distinct losses, followed by a third step of computing eigenpurposes. The natural next step is to investigate how useful these features are for exploration, for instance by learning options which treat these features as rewards, and see if true reward can be accumulated faster than by random exploration.

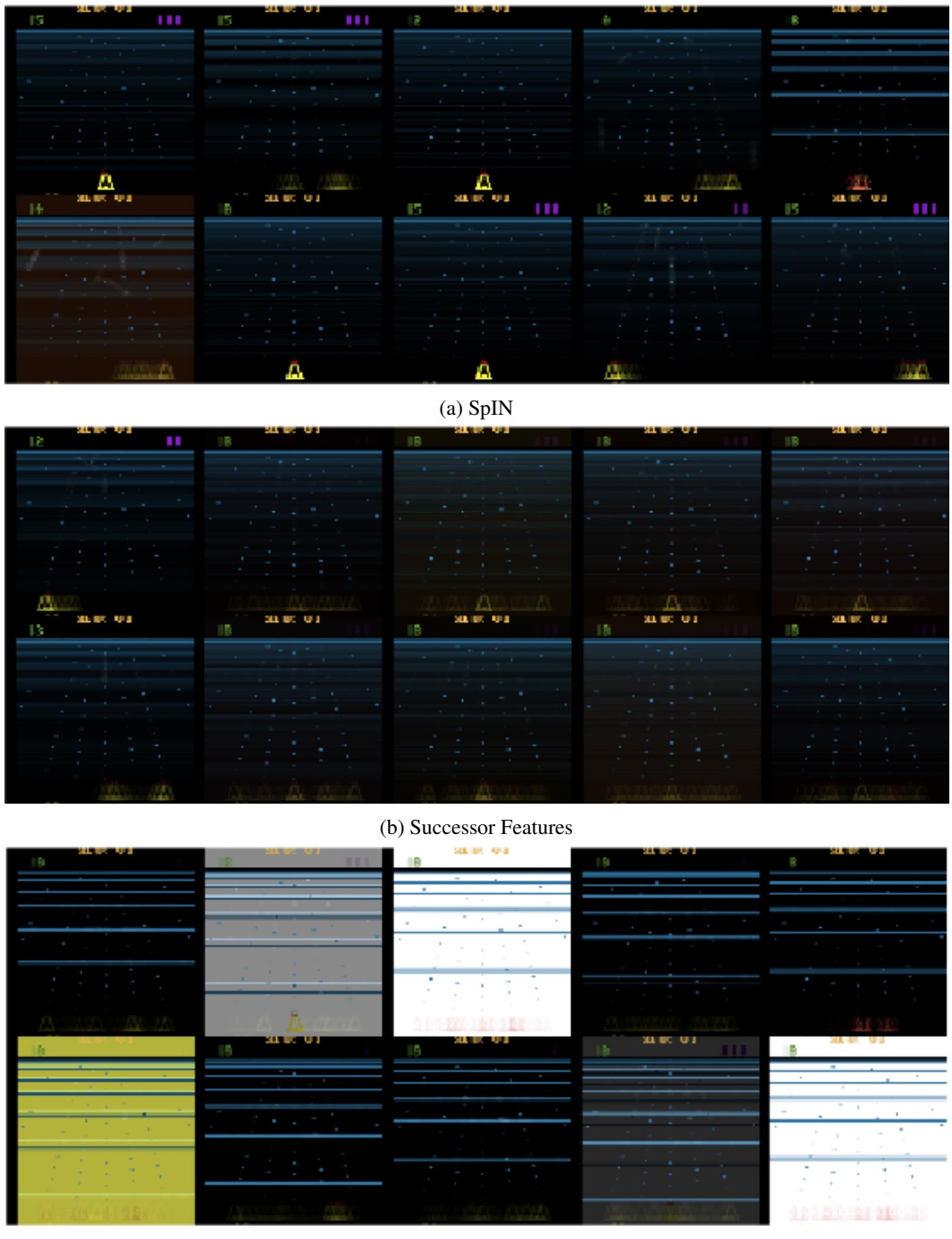

(a) SpIN

(b) Successor Features

(c) PCA

Figure 5: Beam Rider

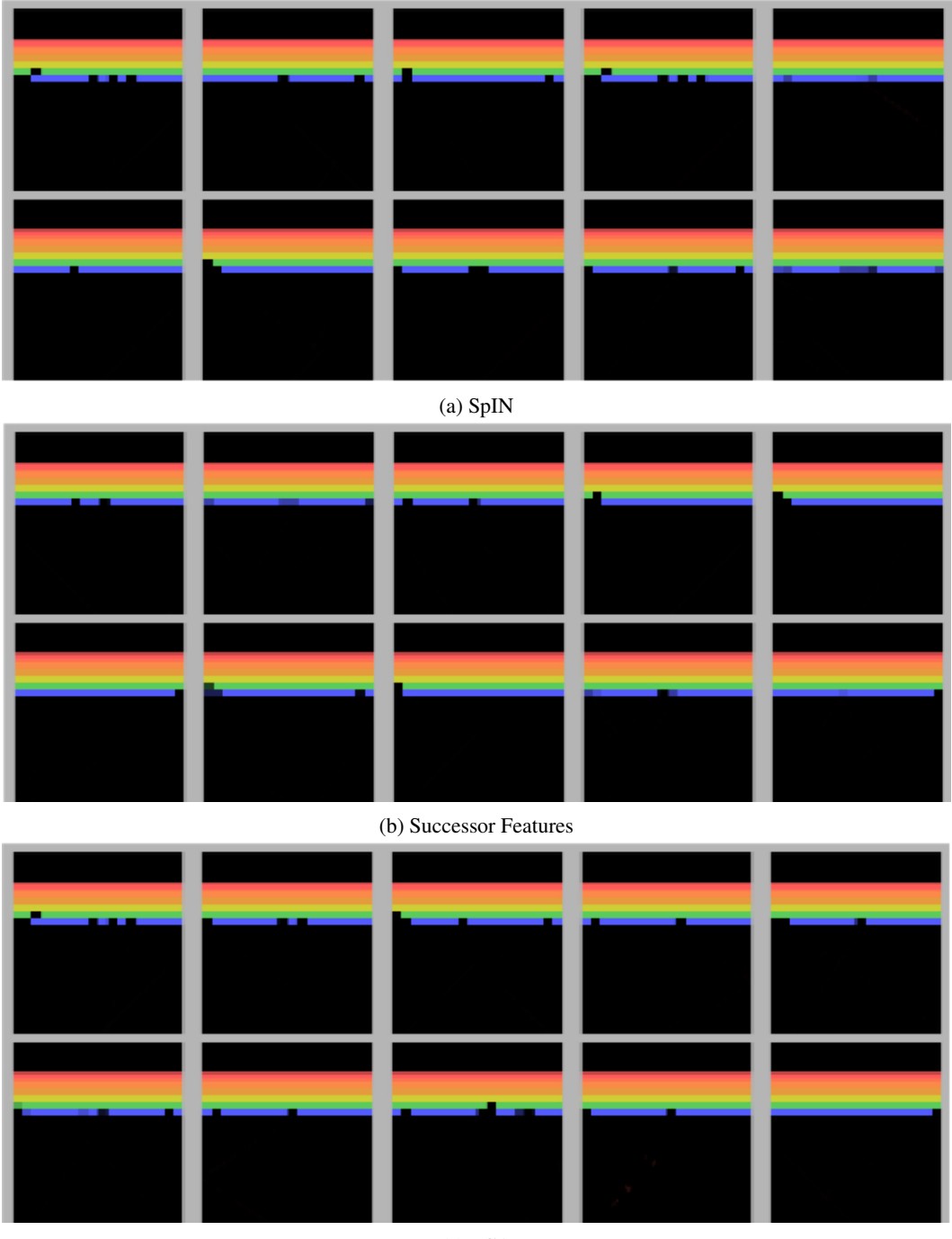

(a) SpIN

(b) Successor Features

(c) PCA

Figure 6: Breakout

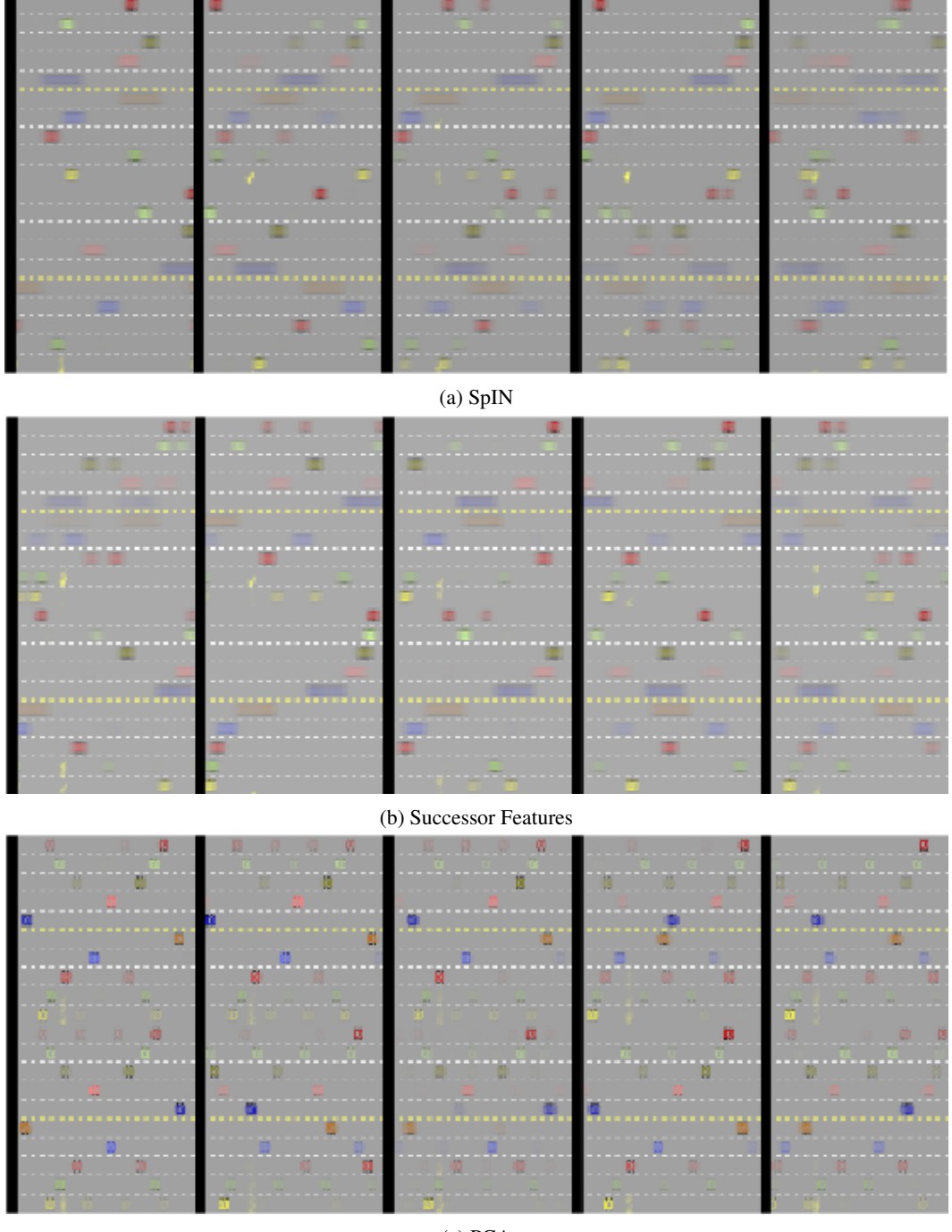

(a) SpIN

(b) Successor Features

(c) PCA

Figure 7: Freeway

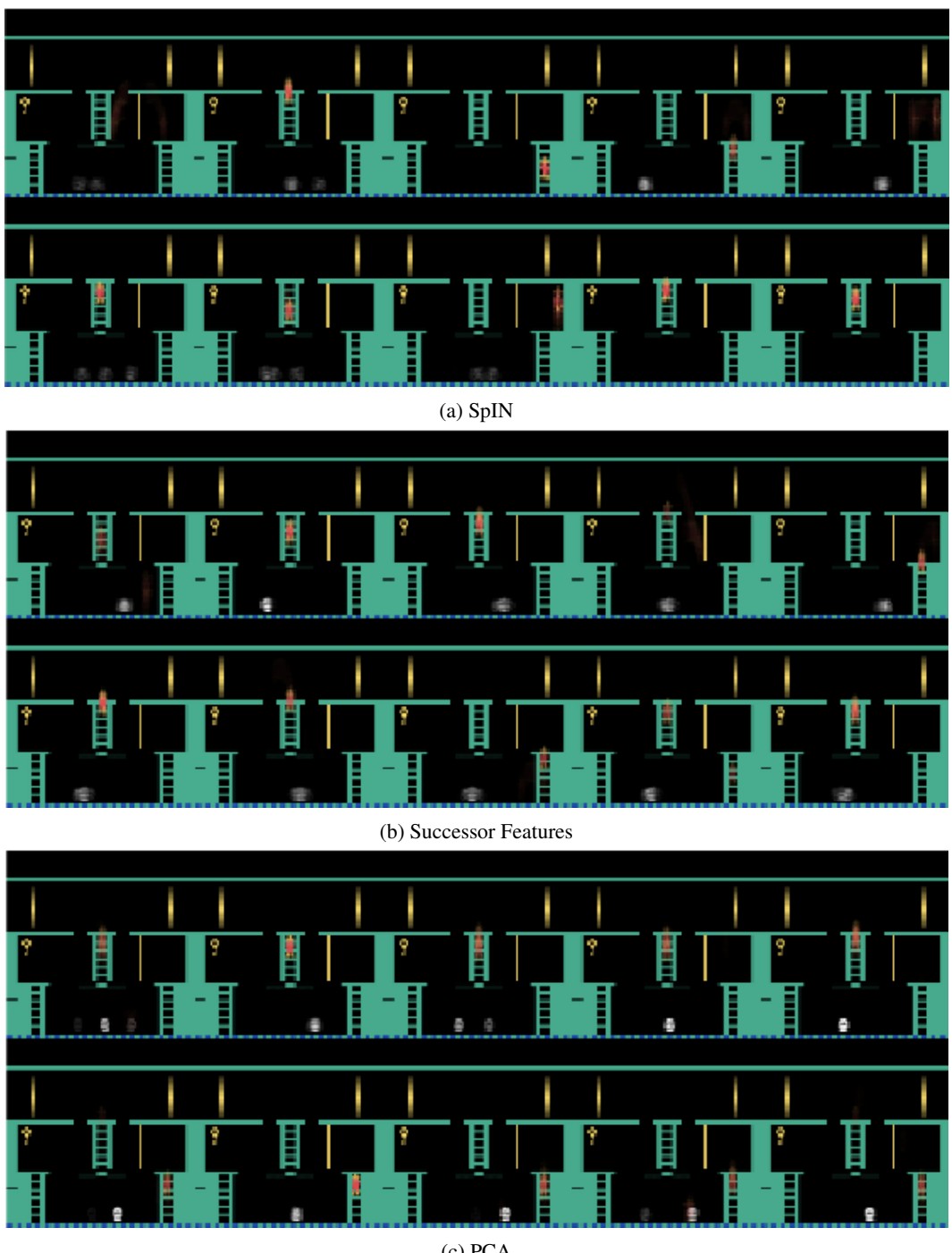

(a) SpIN

(b) Successor Features

(c) PCA

Figure 8: Montezuma's Revenge

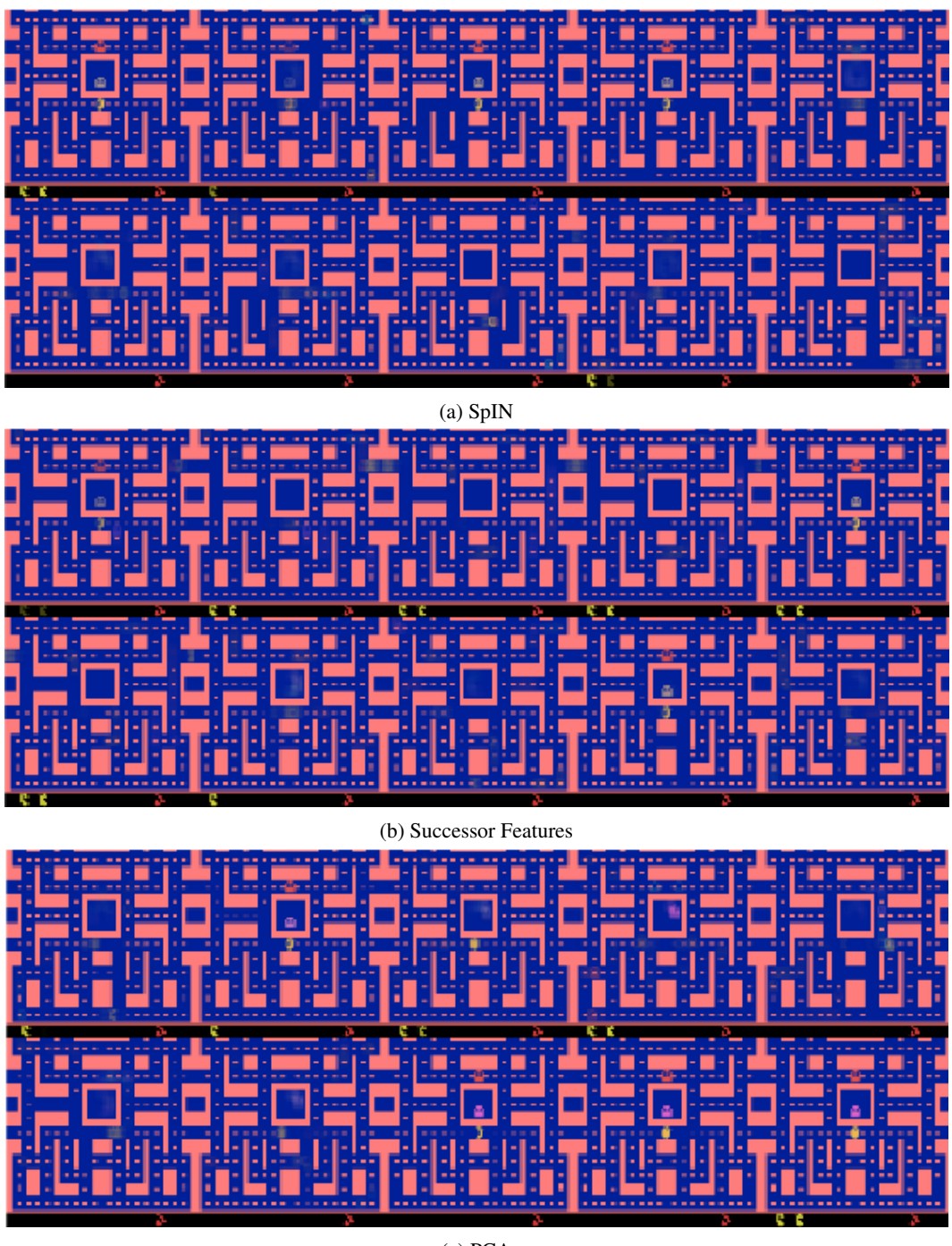

(a) SpIN

(b) Successor Features

(c) PCA

Figure 9: Ms. PacMan

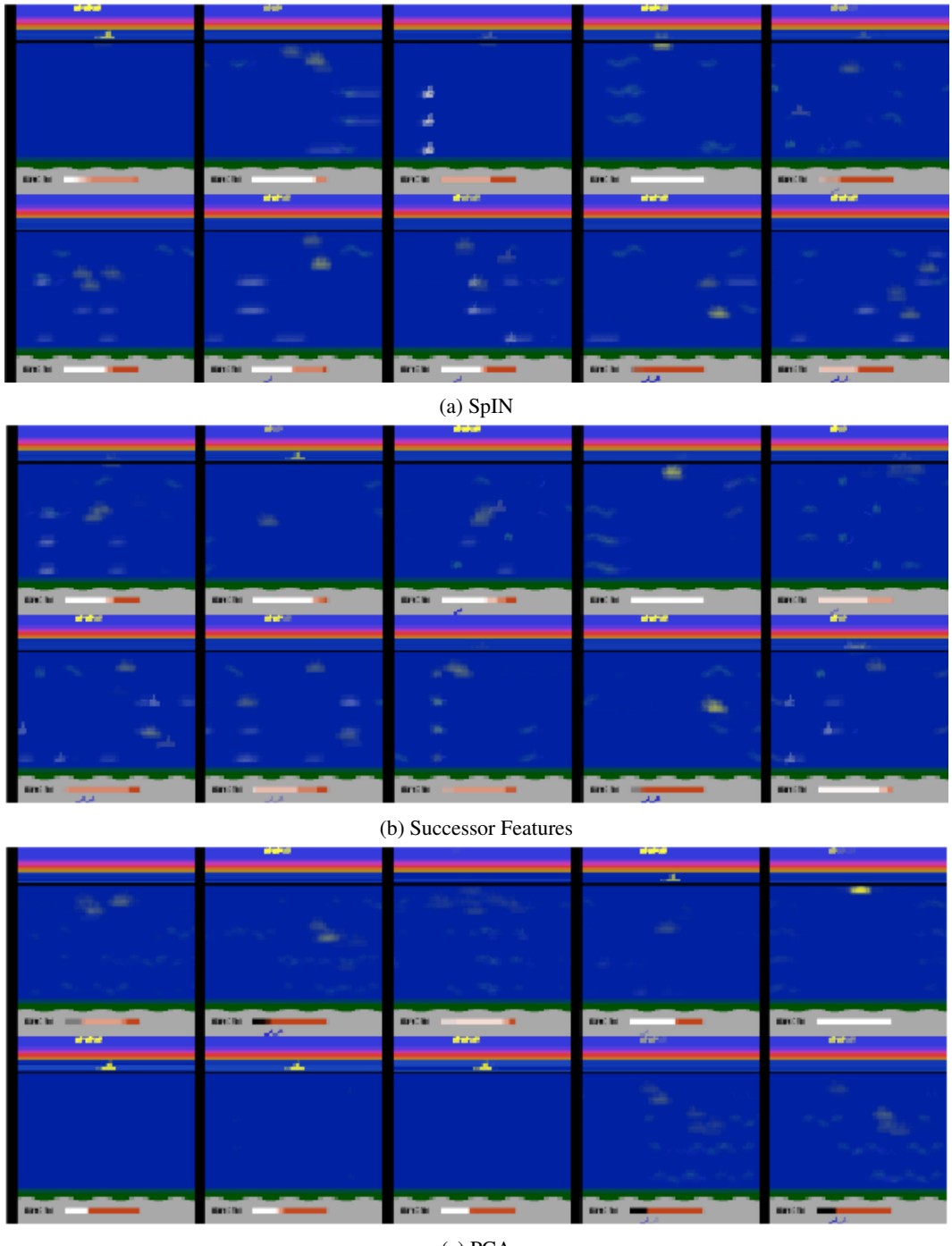

(a) SpIN

(b) Successor Features

(c) PCA

Figure 10: Seaquest

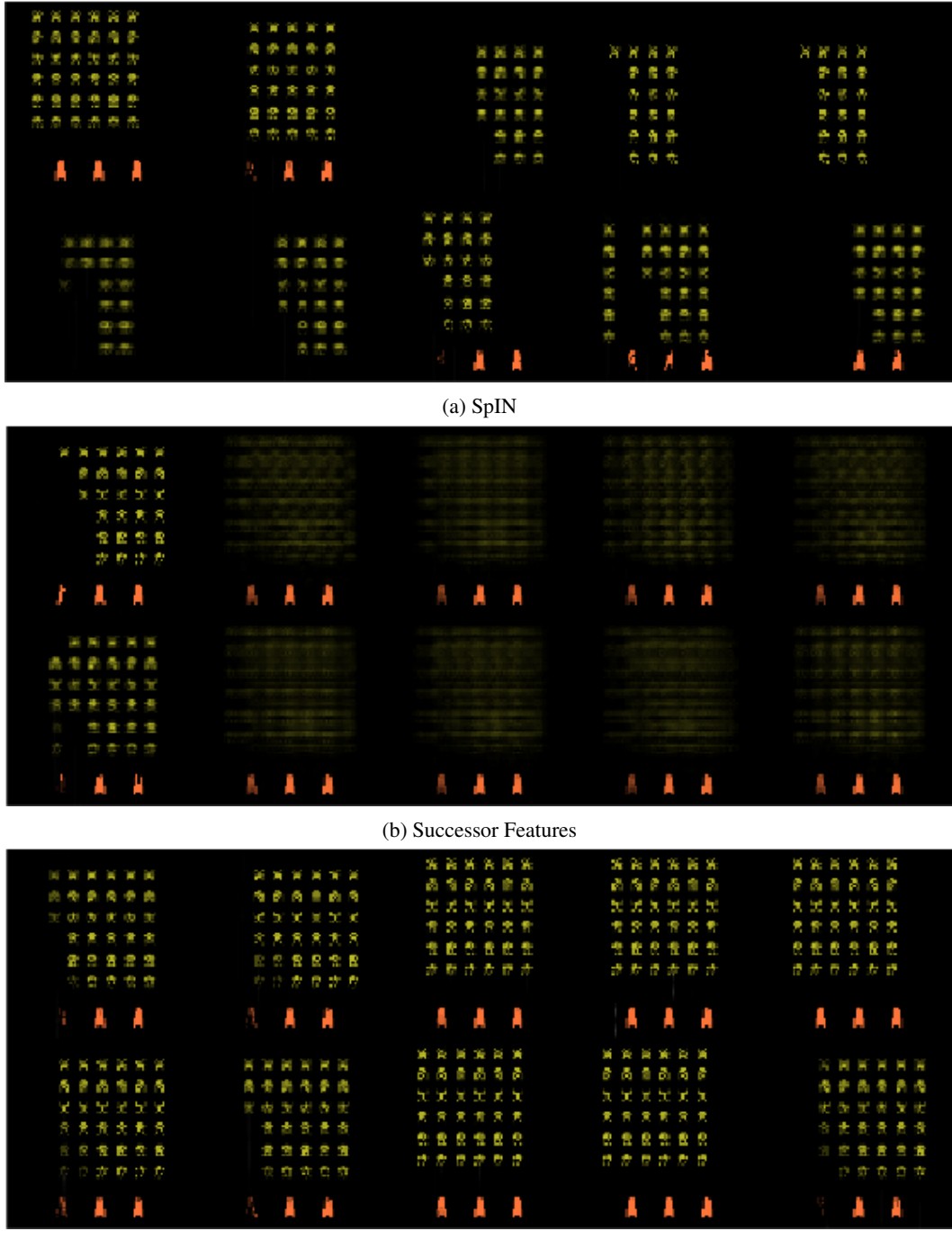

(a) SpIN

(b) Successor Features

(c) PCA

Figure 11: Space Invaders

