# OpenReview forum: "Spectral Inference Networks: Unifying Deep and Spectral Learning"
_ICLR.cc/2019/Conference_

### Official Review · AnonReviewer2 · 2018-11-03
**Good work, but bad presentation**

**Rating:** 5
**Confidence:** 3

**Review:**

In this paper, the authors proposed a unified framework which computes spectral decompositions by stochastic gradient descent. This allows learning eigenfunctions over high-dimensional spaces and generating to new data without Nystrom approximation. From technical perspective, the paper is good. Nevertheless, I feel the paper is quite weak from the perspective of presentation. There are a couple of aspects the presentation can be improved from.

(1) I feel the authors should formally define what a Spectral inference network is, especially what the network is composed of, what are the nodes, what are the edges, and the semantics of the network and what's motivation of this type of network.

(2) In Section 3, the paper derives a sequence of formulas, and many of the relevant results were given without being proven or a reference. Although I know the results are most likely to be correct, it does not hurt to make them rigorous. There are also places in the paper, the claim or statement is inclusive. For example, in the end of Section 2.3, "if the distribution p(x) is unknown, then constructing an explicitly orthonormal function basis may not be possible". I feel the authors should avoid this type of handwaving claims.

(3) The authors may consider summarize all the technical contribution in the paper.

One specific question:

What's Omega above formula (6)? Is it the support of x? Is it continuous or discrete? Above formula (8), the authors said "If omega is a graph". It is a little bit confusing there.

---

> ### Author Response · Authors · 2018-11-20
> **Thank you for your comments**
>
> Response: We thank the reviewer for their comments. We are glad that they found the technical contribution strong, and hope we can address their issues with the presentation. To the specific points raised:
>
> (1) We use the term “network” in spectral inference networks in the sense of “neural network”, similar to “generative adversarial networks” or “Hopfield networks”. The nodes and edges would be exactly the same as for any other neural network architecture, and we describe the exact network architectures used in the supplementary materials in Section C. As these were fairly conventional network architectures, we did not want to use the already tight space in the main paper to describe them. To reiterate the point we made to Reviewer 1 - we are happy to move these details into the main paper, but it will put us over 8 pages. If this is a deciding factor in raising the score, we’ll do it.
>
> As for what the defining characteristics of a spectral inference network are, as opposed to other neural network architectures or machine learning frameworks, there are three key ingredients:
> * The loss in Eq 6
> * The symmetry-broken gradient in Eq 14, which provides a natural ordering to the output of the network
> * The use of moving averages of the covariance and Jacobian of the covariance (line 7 and 8 of Alg 1) to correct for the bias in the gradients with bilevel optimization
> We summarize this training algorithm in Alg 1, but will include it in the text as well.
>
> (2) The full derivation of the expressions in Section 3 are too long to fit in the body of an already-tight paper. We provide a step-by-step derivation of every relevant expression in Section 3 in the supplementary material in Section A, and we provide references for all other derivations.
>
> As for constructing an explicitly orthonormal basis without p(x) being known - we feel it is a self-evident statement that one cannot construct an orthonormal basis in closed form with respect to an inner product which is not known. We have rewritten this and the following statement to make it more concrete. If there are any other places in the paper which similarly could be improved, please let us know.
>
> (3) We’re not entirely sure how to respond to this. We feel we’ve described the technical contribution of the paper quite well. Perhaps you could give a more concrete example of how you feel we could improve or what you think is missing?
>
> And to your specific question: yes, Omega is the support of x. It can be both continuous (as in the hydrogen atom example, where Omega is R^2) or discrete, as in any case with graph-structured data. The only requirement on Omega is that it is a measurable space. We have clarified this in the paper.

---

### Official Review · AnonReviewer1 · 2018-11-03
**large-scale spectral decomposition - high practical value**

**Rating:** 7
**Confidence:** 3

**Review:**

Spectral Inference Networks, Unifying Deep and Spectral Learning

This paper presents a framework to learn eigenfunctions via a stochastic process. They are exploited in an unsupervised setting to learn representation of video data. Computing eigenfunctions can be computationally challenging in large-scale context. This paper proposes to tackle this challenge b y approximating then using a two-phase stochastic optimization process. The fundamental motivation is to merge approaches from spectral decomposition via stochastic approximation and learning an implicit representation. This is achievement with a clever use of masked gradients, Cholesky decomposition and explicit orthogonalization of resulting eigenvectors. A bilevel optimization process finds local minima as approximate eigenfunction, mimicking Borkar’97. Results are shown to correctly recover known 2d- schrodinger eigenfunctions and interpretable latent representation a video dataset, with a practical promising results using the arcade learning environment.

Positive
+ Computation of eigenfunctions on very large settings, without relying on Nystrom approximation
+ Unifying spectral decomposition within a neural net framework

Specific comments
- Accuracy issue - Shape of eigenfunctions are said to be correctly recovered, but no words indicates their accuracy. If eigenfunction values are wrong, this may be critical to the generalization of the method.
- Clarity could be improved in the neural network implementation, what is exactly done and why, when building the network
- Algorithm requires computing the jacobian of the covariance, which can be large and computationally expensive - how to scale it to large settings?
- Fundamentally, a local minimum is reached - any future work on tackling a global solution?  Perhaps by exploring varying learning rates?
- Practically, eigenfunction have an ambiguity to rotation - how is this enforced and checked during validation? (e.g., rotating eigenfunctions in Fig 1c)
- Eigenfunction of transition matrix should, if not mistaken, be smooth, whereas Fig 2a shows granularity in the eigenfunctions values (noisy red-blue maps) - Is this regularization issue, and can this be explicitly correctly?
- Perhaps a word on computational time/complexity?

---

> ### Author Response · Authors · 2018-11-20
> **Thank you for your comments**
>
> Thank you for your kind words and comments, and we are gratified that you recognize the high potential for practical impact of our work. To the specific criticisms and suggestions you mention:
>
> Accuracy: We believe that any significant inaccuracy in the shape of the learned eigenfunctions for the hydrogen atom would be reflected in the energy. For instance, if the learned solution was not smooth enough, the Laplacian term in the Hamiltonian would be too high, and this would have a noticeable effect on the loss. The fact that the loss converges to close to the known closed form solution gives us good confidence in the accuracy of the method. We have also done follow-up experiments since the initial submission that achieve even higher accuracy on the energy, but feel that these additional experiments are outside the scope of this paper .
>
> Clarity: We apologize if any details were unclear. We saved the details of the network architecture for the supplementary materials, but if anything in Section C was unclear or insufficient please let us know and we’ll correct it. We are also happy to move the network architecture details into the main body of the paper. This would put the paper over 8 pages, but if you feel it would significantly improve the quality we’ll go ahead and do it. Since the focus of this paper was on the loss function and optimization procedure for spectral inference networks, we put less emphasis on choosing a network architecture, and made mostly conventional choices in our network design.
>
> Scaling: You are correct to point out that computing the Jacobian of the covariance of the features is the bottleneck of this approach. We believe that any strong paper should be as honest about the weaknesses of the proposed approach as the strengths, and we are sure to point out at the end of section 3.4 exactly what you mentioned. Out of all the ways we tried to approximate the gradient of the Rayleigh quotient in time for the submission deadline, using a moving average of the Jacobian was the stablest and fastest to converge. Since submission, we have done significant work on alternatives that scale better, and believe we have some promising candidates, but feel that this is best left to a future publication, since it constitutes a significant body of additional material.
>
> Local minima: We would love to be able to achieve a global minimum - but the fact that we are reaching a local rather than global minimum is entirely because we use neural networks as a function approximator. If we could guarantee global convergence of neural networks on any problem it would be a much bigger deal than just improving spectral learning!
>
> Ambiguity: You are correct that there is an ambiguity in the eigenfunctions *if there are degenerate eigenfunctions*, that is, if there are two or more eigenfunctions with identical eigenvalues. This is in fact the case in the 2D hydrogen atom. The degenerate solutions take the form of different spherical harmonics, so as long as the solutions we find look recognizably like spherical harmonics (i.e. rotated versions of the solutions found in Fig 1a), and the energies are correct, then we are confident in our results.
>
> Smoothness: What is going on in Fig 2a is not a perfect visualization of the eigenfunctions. The true underlying state space for the video is 12 dimensional (2 position, 2 momentum, 3 balls) with some symmetry due to the indistinguishability of the balls. We are taking that 12 dimensional state space and projecting it down to 2 dimensions, as well as mixing different dimensions together, because each frame of the video contributes 3 points to the visualization (one for each ball). Plotting the position of all 3 balls on the same 2D space is most likely what gives the figures the “speckled” look. We will include this additional explanation in the paper.
>
> Computational complexity: We already briefly touch on the complexity of computing the Jacobian in section 3.4, and we believe that the convergence results for two-time-scale optimization are not so different from the standard results in stochastic optimization (i.e. 1/sqrt(T) convergence rate). However we can add more detail in the paper explaining this.
>
> Once again, we’re very glad that you enjoyed the paper, and thank you for the comments on how to improve it even more!

---

### Official Review · AnonReviewer3 · 2018-11-05
**linear algebra with deep learning framework (Tensorflow)**

**Rating:** 5
**Confidence:** 3

**Review:**

In this paper, the authors propose to use a deep learning framework to solve a problem in linear algebra, namely the computation of the largest eigenvectors.

I am not sure tu understand the difference between the framework described in sections 3.1 and 3.2. What makes section 3.2 more general than 3.1?
In particular, the graph example in section 3.2 with the graph Laplacian seems to fit in the framework of section 3.1. What is the probability p(x) in this example? Similarly for the Laplace-Beltrami operator what is the p(x)? I do not understand the sentence: 'Since these are purely local operators, we can replace the double expectation over x and x' with a single expectation.'

The experiments section is clearly not sufficient as no comparison with existing algorithms is provided. The task studied in this paper is a standard task in linear algebra and spectral learning. What is the advantage of the algorithm proposed in this paper compared to existing solutions? The authors provide no theoretical guarantee (like rate of convergence...) and do not compare empirically their algorithm to others.

---

> ### Author Response · Authors · 2018-11-20
> **Thank you for your comments**
>
> We thank the reviewer for their comments. To the first comment, on the distinction between section 3.1 and 3.2 - section 3.1 deals only with the case of functions on finite, discrete spaces (that is, vectors). This is to ease the reader into the discussion of linear operators and eigenfunctions from the more familiar point of view of matrices and eigenvectors. Once the reader understands how eigenvectors can be derived as the solution to an optimization problem, the extension to arbitrary function spaces should be easier.
>
> The discussion in section 3.2 pertains to *all* measurable spaces: discrete, continuous, compact or unbounded. We have updated the paper to clarify this. If we were writing the integral for <f, g> in section 3.2 in more formal measure-theoretic notation, we would express it as an integral over some measure d\mu instead of p(x)dx. This measure could include the uniform measure, which generalizes uniform distributions to spaces with infinite total measure. In this case p(x) would just be a constant. We tried to highlight this point without burdening the reader with too much formal measure theory where we say “In theory this could be an improper density, such as the uniform distribution over R^n”.
>
> The section on the graph Laplacian and Laplace-Beltrami operator is not really specific to section 3.2 or functions on continuous spaces. Rather we wanted to shift the paper to a more general discussion of the types of kernels that might appear in different spectral problems. We can break this off into a separate section 3.3 to avoid confusion, but that will put the paper over 8 pages. If you think such a change is worth the paper going longer and would be the deciding factor in raising your score, we will happily do it.
>
> The Laplacian (either graph or manifold) is really a specific choice of kernel k(x, x’), which can then be plugged in with any p(x) to define a linear operator. Depending on the application, common choices of p(x) would be the data distribution (for machine learning application) or the uniform distribution (i.e. a constant). When we say the Laplacian in continuous space is a local operator, we mean that the value of K[f](x) depends solely on the value of f and its first and second order derivative at x. Again, we will rewrite this section to make this more clear.
>
> To the point on comparisons against the state of the art - the aim of this paper was to show we could compute meaningful spectral decompositions *at a scale larger than any existing method*. We don't provide quantitative comparisons to other methods because our method scales to solve problems that are orders of magnitude more difficult than the most difficult problems that standard methods can address. As we state in paragraph two of the introduction, using an existing method like the Nystrom approximation for generalization “is not practical for large datasets, and some form of function approximation is necessary.” For a sense of the scale at which exact spectral methods become impractical, please take a look at Perozzi, Al-Rfou and Skiena, KDD 2014. There they are unable to run spectral clustering on the YouTube dataset, which consists of a graph with over 1 million nodes and nearly 3 million edges - a scale which SpIN can easily handle. Neither their proposed algorithm, nor any of the other scalable baselines, are a true spectral method.
>
> Spectral inference networks are especially powerful in the case of large datasets *and high dimensional data*, as any neural network architecture can be applied as an eigenfunction approximator. This was why we chose the example of videos of bouncing balls as the second experiment. Existing spectral methods do not scale to this type of data. We also compare our algorithm against an approach to approximate spectral learning used by Machado et al in ICLR 2017 in section C.3 of the supplementary material. In that paper they claimed to learn eigenfunctions of the successor operator in reinforcement learning environments. However the approximate eigenfunctions they learn have no clear ordering as you would expect from true eigenfunctions. By contrast, the eigenfunctions learned by spectral inference networks are clearly more meaningful and learn features that are more distinguishable, even by the naked eye. It is known that the eigenfunctions of the successor operator can be useful for reinforcement learning tasks.

---

### Meta-Review · Area_Chair1 · 2018-12-12
**Some presentation issues, but practical value for large-scale eigen computations**

**Confidence:** 4
**Recommendation:** Accept (Poster)

**Metareview:**

The paper proposes a deep learning framework to solve large-scale spectral decomposition.

The reviewers and AC note that the paper is quite weak from presentation. However, technically, the proposed ideas make sense, as Reviewer 1 and Reviewer 2 mentioned. In particular, as Reviewer 1 pointed out, the paper has high practical value as it aims for solving the problem at a scale larger than any existing method. Reviewer 3 pointed out no comparison with existing algorithms, but this is understandable due to the new goal.

In overall, AC thinks this is quite a boarderline paper. But, AC tends to suggest acceptance since the paper can be interested for a broad range of readers if presentation is improved.